# A Quantitative Approach to Predicting Representational Learning and Performance in Neural Networks

## Abstract

A key property of neural networks (both biological and artificial) is how they learn to represent and manipulate input information in order to solve a task. Different types of representations may be suited to different types of tasks, making identifying and understanding learned representations a critical part of understanding and designing useful networks. In this paper, we introduce a new pseudo-kernel based tool for analyzing and predicting learned representations, based only on the initial conditions of the network and the training curriculum. We validate the method on a simple test case, before demonstrating its use on a question about the effects of representational learning on sequential single versus concurrent multitask performance. We show that our method can be used to predict the effects of the scale of weight initialization and training curriculum on representational learning and downstream concurrent multitasking performance.

## 1 Introduction

One of, if not the, most fundamental question in neural networks research is how representations are formed through learning. In machine learning, this is important for understanding how to construct systems that learn more efficiently and generalize more effectively (Bengio et al., 2013; Witty et al., 2021). In cognitive science and neuroscience, this is important for understanding how people acquire knowledge (Rumelhart et al., 1993; Rogers & McClelland, 2004; Saxe et al., 2019), and how this impacts the type of processing (e.g., serial and control-dependent vs. parallel and automatic) used in performing task(s) (Musslick & Cohen, 2021; Musslick et al., 2020). One important focus of recent work has been on the kinds of inductive biases that influence how learning impacts representations (e.g., weight initialization, regularization in learning algorithms, etc. (Narkhede et al., 2022; Garg & Liang, 2020)) as well as training curricula (Musslick et al., 2020; Saglietti et al., 2022). This is frequently studied using numerical methods, by implementing various architectural or learning biases and then simulating the systems to examine how these impact representational learning (Caruana, 1997; Musslick et al., 2020). Recently, Sahs et al. (2022) introduced a novel analytic approach to this problem, which can be used to predict important inductive bias properties from the network's initialization. Here we extend this approach by combining it with a neural tangent kernel-based analysis in order to qualitatively predict the kinds of representations that are learned, and the consequences this has for processing. We provide an example that uses a neural network model to address how people acquire simple tasks, and the extent to which this leads to serial, control-dependent versus parallel, automatic processing and multitasking capability (Musslick et al., 2016; Musslick & Cohen, 2021; Musslick et al., 2020). We expand upon theoretical results introduced in Sahs et al. (2022), combining them with a neural tangent kernel analysis (Jacot et al., 2018), which allows for prediction of the inductive bias (and resulting representations learned by the network and downstream task performance) from the initial conditions of the network and the training regime. In the remainder of this section, we provide additional background that motivates the example we use. Then, in the sections that follow, we describe the analysis method, its validation in a benchmark setting, and the results of applying it to a richer and more complex example.

*Shared versus separated representations and flexibility versus efficiency.* One of the central findings from machine learning research using neural networks is that cross-task generalization (sometimes referred to as

transfer learning) can be improved by manipulations that promote the learning of shared representations—that is, representations that capture statistical structure that is shared across tasks (Caruana, 1997; Baxter, 1995; Collobert & Weston, 2008). One way to do so is through the design of appropriate training regimens and/or learning algorithms (e.g., multi-task learning and/or meta-learning; (Caruana, 1997; Ravi et al., 2020)). Another is through the initialization of network parameters; for example, it is known that small random initial weights help promote the learning of shared structure, by forcing the network to start with what amounts to a common initial representation for all stimuli and tasks and then differentiate the representations required for specific tasks and/or stimuli under the pressure of the loss function (Flesch et al., 2021). Interestingly, while shared representations support better generalization and faster acquisition of novel but similar tasks, this comes at a cost of parallel processing capacity, a less commonly considered property of neural networks that determines how many distinct tasks the system can perform *at the same time*—that is, its capacity for concurrent multitasking (Feng et al., 2014; Musslick et al., 2016; Petri et al., 2021b). Note that our use of the term "multitasking" here should not be confused with the term "multitask" learning: the former refers to the simultaneous *performance* of multiple tasks, while the latter refers to the simultaneous acquisition of multiple tasks. These are in tension: if two tasks share representations, they risk making conflicting use of them if the tasks are performed at the same time (i.e., within a single forward-pass); thus, the representations can be used safely only when the tasks are executed serially. This potential for conflict can be averted if the system uses *separate* representations for each task, which is less efficient but allows multiple tasks to be performed in parallel. This tension between shared vs. separated representations reflects a more general tradeoff between the *flexibility* afforded by shared representations (more rapid learning and generalization) but at the expense of serial processing, and the *efficiency* afforded by separated, task-dedicated representations (parallel processing; i.e., multitasking) but at the costs of slower learning and poorer generalization (i.e., greater rigidity; Musslick & Cohen (2021); Musslick et al. (2020)). While this can be thought of as analogous to the tension between interpreted and compiled procedures in traditional symbolic computing architectures, it has not (yet) been widely considered within the context of neural network architectures in machine learning.

*Shared versus separated representations and control-dependent versus automatic processing.* The tension between shared and separated representations also relates to a cornerstone of theory in cognitive science: the classic distinction between control-dependent and automatic processing (Posner & Snyder, 1975; Shiffrin & Schneider, 1977). The former refers to "intentional," "top-down," processes that are assumed to rely on control for execution (such as mental arithmetic, or searching for a novel object in a visual display), while the latter refers to processes that occur with less or no reliance on control (from reflexes, such as scratching an itch, to more sophisticated processes such as recognizing a familiar object or reading a word). A signature characteristic of control-dependent processes is the small number of such tasks that humans can perform at the same time—often only one—in contrast to automatic processes that can be performed in parallel (motor effectors permitting). The serial constraint on control-dependent processing has traditionally been assumed to reflect limitations in the mechanism(s) responsible for control *itself*, akin to the limited capacity imposed by serial processing in the core of a traditional computer (Anderson & Lebiere, 2014; Pashler, 1994; Posner & Snyder, 1975). However, recent neural network modeling work strongly suggests an alternative account: that constraints in control-dependent processing reflect the imposition of serial execution on processes that rely on shared representations (Musslick & Cohen, 2021; Musslick et al., 2020). That is, constraints associated with control-dependent processing reflect the *purpose* rather than an intrinsic *limitation* of control mechanisms. This helps explain the association of control with flexibility of processing Cohen (2017); Duncan (2001); Goschke (2000); Kriete et al. (2013); Shiffrin & Schneider (1977); Verguts (2017): flexibility is afforded by shared representations, which require control to insure they are not subject to conflicting use by competing processes. It also explains why automaticity—achieved through the development of task-dedicated representations—takes longer to acquire and leads to less generalizable behavior (Logan, 1997). Together, these explain the canonical trajectory of skill acquisition from dependence on control to automaticity: When people first learn to perform a novel task (e.g., to type, play an instrument, or drive a car) they perform it in a serial, control-dependent manner, that precludes multitasking. Presumably this is because they exploit existing representations that can be "shared" to perform the novel task as soon as possible, but at the expense of dependence on control. However, with extensive practice, they can achieve efficient performance through the development of separated, task-dedicated representations that diminish

reliance on control and permit performance in parallel with other tasks (i.e., concurrent multitasking; Garner & Dux (2015); Musslick & Cohen, J. D. (2019)).

These ideas have been quantified in mathematical analyses and neural network models, and fit to a wide array of findings from over half a century of cognitive science research (Musslick et al., 2020). However, the specific conditions that predispose to, and regulate the formation of shared versus separated representations are only qualitatively understood, and theoretical work has been restricted largely to numerical analyses of learning and processing in neural network models. Some methods have sought to quantify the degree of representation sharing between two tasks in terms of correlations between activity patterns for individual tasks (Musslick et al., 2016; 2020; Petri et al., 2021a;b) in order to predict multitasking capability. Other methods, that quantify the representational manifold of task representations (Bernardi et al., 2020) have been applied to characterize multitasking capability (Henselman-Petrusek et al., 2019). However, while these methods provide a snapshot of representation sharing at a given point in training, they do not provide direct or analytic insight into the *dynamics* of learning shared versus separated representations, nor how the inductive bias of a system may affect the representations learned.

In this article, we expand upon the ideas introduced in Sahs et al. (2022) to show how the initial condition of a network and the training regime to which it will be subjected can predict the implicit bias and thus the kinds of representations it will learn (e.g., shared vs. separated) and the corresponding patterns of performance it will exhibit in a given task setting. This offers a new method to analyze how networks can be optimized to regulate the balance between flexibility and efficiency. The latter promises to have relevance both for understanding how this is achieved in the human brain, and for the design of more adaptive artificial agents that can function more effectively in complex and changing environments.

*Task structure and network architecture.* For the purposes of illustration and analysis, we focus on feedforward neural networks with three layers of processing units that were trained to perform sets of tasks involving simple stimulus-response mappings. Each network was comprised of an input layer, subdivided into pools of units representing inputs along orthogonal stimulus dimensions (e.g., representing colors, shapes, etc.), and an additional pool used to specify which task to perform. All of the units in the input layer projected to all of the units in the hidden layer, which all projected to all units in the output layer, with an additional projection from the task specification input pool to the output layer. The output layer, like the input layer, was divided into pools of units, in this case representing outputs along orthogonal response dimensions (e.g., representing manual, verbal, etc.).

Networks were trained in an environment comprised of several feature groups (e.g., shape, size, etc.) and response groups (e.g., verbal, manual, etc.) corresponding to the stimulus and response dimensions along which the pools of input and output units of the networks were organized. Each network was trained to perform a set of tasks, in which each task was defined by a one-to-one mapping from the inputs in one pool (i.e., along one stimulus feature dimension) to the outputs in a specified pool (i.e., along one response dimension), ignoring inputs along all of the other feature dimensions and requiring null outputs along all of the other response dimensions.[1] Training and testing could be performed for one task at a time ("single task" conditions), by specifying only that task in the task input pool and requiring the correct output over the task-relevant response dimension and a null response over all others; or with two or more tasks in combination ("multitasking" conditions), in which the desired tasks were specified over the task input units, and the network was required to generate correct responses over the relevant response dimensions and null responses for all others. In all cases, an input was always provided along every stimulus dimension, and the network had to learn to ignore those that were not relevant for performing the currently specified task(s). In each case, the question of interest was how initialization and learning impacted the final connection weights to and from the hidden layer, the corresponding representations the networks used to perform each task, and the patterns of performance in single task and multitasking conditions. Specifically, we were interested in the extent to which the networks learned shared versus separated representations for sets of tasks that shared a common feature dimension; and the extent to which the analytic methods of interest were able to predict, from the initial conditions and task specifications, the types of representations learned, and the corresponding patterns of performance (e.g., speed of learning and multitasking capability). We evaluated

---

[1]This corresponds to the formal definition of tasks in a task space as described in Musslick et al. (2020).

the evolution of representations over the course of learning in two ways: using the analytic techniques of interest, and using a recently developed visualization tool to inspect these, each of which we describe in the two sections that follow.

## 2 Understanding and Visualizing Learning Dynamics

### 2.0.1 Notation

We parameterize a neural network as a function $f$, which takes in inputs $x$ and targets $y$ and generates responses $\hat{y} = f(x)$. Our train set is $m \in M$, while our test set is $n \in N$, such that $x_m$ is the $m$th training input. Our NN is assumed to be trained via a variant of gradient descent, with learning rate $\eta$. Our NN is parameterized by a vector of $P$ parameters, $\theta \in \mathcal{R}^P$. The NN is also trained using a loss function $L(\hat{y}, y)$.

### 2.1 The empirical Neural Tangent Kernel (eNTK)

Understanding the learning dynamics of a neural network (NN) can be done using a framework known as the Neural Tangent Kerel (NTK). We use a related quantity we call the empirical NTK (eNTK).

The NTK is based on a kernel function $K(x, x')$ that represents the 'similarity' of inputs $x$ and $x'$ (from here on, we use $x$ from the training set and $x'$ from the test set); that is, how much influence each individual sample $x$ from the training set has on the output decision of the NN on a test sample $x'$. This is embodied in a kernel expansion (Jacot et al., 2018), which uses an assumption that the width of the network's layers goes to infinity in order to use a Gaussian-process based simplification. Empirical work with finite-width NTKs has led to interesting use cases, including analyzing adversarial training Loo et al. (2022) or analysis of learning trajectories and loss landscapes Fort et al. (2020); Lewkowycz et al. (2020). In this work, we offer a slightly different derivation more suited to the eNTK:

Using gradient descent (GD) on a NN with scalar learning rate $\eta$ and a $P \times 1$ vector of real parameters $\theta$, the parameter update can be written as

$$\theta(t + 1) = \theta(t) - \eta \frac{dL(\theta)}{d\theta}. \tag{1}$$

Taking the gradient flow approximation $\eta \to 0$ (e.g. as the step size approaches 0, resulting in a continuous flow rather than discrete steps) we have that the rate of parameter change is

$$\frac{d\theta}{dt} = -\frac{dL(\theta)}{d\theta} \tag{2}$$

Assuming our loss depends only on the network output $\hat{y}$, we can rewrite this as a sum over $N$ training samples $\mathcal{D} := \{(x_m, y_m)\}$:

$$\frac{d\theta}{dt} = -\frac{dL}{d\hat{y}} \frac{d\hat{y}}{d\theta} = \sum_{m \in \mathcal{D}} \frac{dL}{d\hat{y}_m} \frac{d\hat{y}_m}{d\theta} := \sum_{m \in \mathcal{D}} \epsilon_m \phi_m, \tag{3}$$

where we have defined loss sensitivity $\epsilon_m \in \mathbb{R} = \frac{dL}{d\hat{y}_m}$ and feature vector $\phi_m \in \mathbb{R}^P = \frac{d\hat{y}_m}{d\theta_p}$ for each parameter $p$ for a given sample $x_m$. How does the actual NN predicted output ($\hat{y}$) change with learning over time? We can answer this by taking total time derivatives yielding

$$\frac{d\hat{y}}{dt} = \frac{d\hat{y}(\theta)}{d\theta}^T \frac{d\theta}{dt} = -\frac{d\hat{y}(\theta)}{d\theta}^T \frac{dL(\theta)}{d\hat{y}} \frac{d\hat{y}(\theta)}{d\theta} \tag{4}$$

where the eNTK is the weighted kernel function $\epsilon_{x'} K(x, x'; \theta) := \frac{dL(\theta)}{d\hat{y}(x')} \frac{d\hat{y}(x;\theta)}{d\theta}^T \frac{d\hat{y}(x';\theta)}{d\theta}$. Note that this means the network's time evolution is a kernel function, made up of the NTK at time $t$ with parameters $\theta = \theta(t)$ and kernel weights $\epsilon = \frac{dL(\theta)}{d\hat{y}}$ Breaking down equation 4 as a sum over the training set:

$$\frac{d\hat{y}}{dt} = -\sum_{m \in \mathcal{D}} K_{mn}(t) \epsilon_m(t), \quad \forall n \in \mathcal{D} \tag{5}$$

which expresses the evolution of the test example $x_m$ as a weighted kernel function (the NTK) of the train set, where the kernel element $K_{mn}(t)$ is the $[m, n]$ entry of $K$ e.g.

$$\frac{d\hat{y}(x_m, \theta(t))}{d\theta}^T \frac{d\hat{y}(x_n, \theta(t))}{d\theta} = \phi_m^T \phi_n \tag{6}$$

revealing that the kernel feature vector is just $\phi$.

If the model is close to linear in the parameters $\theta$ (i.e., we are in the so-called *kernel* or *lazy* training regime Chizat et al. (2019), where the NN's basis functions are fixed for all time), then the NTK will not change much during training (as in the linear regime $\frac{d\hat{y}}{d\theta}$ is slowly varying), allowing the entire learning to be readily interpretable as a linear kernel machine (Ortiz-Jiménez et al., 2021). In this case, each update to the model is fully interpretable under kernel theory, with each data point influencing how the model evolves (see Fig. 1).

But what if the model is not close to the linear/lazy/kernel regime (i.e., it is in the so-called *adaptive regime*?[2] In this case the NN's basis functions do change over time, rendering the NTK function time-dependent (in which case we denote it as $K(x, x', t)$). In this case, the classical NTK theory no longer holds. However, the eNTK update equation can easily be computed at any time t.

The eNTK and resulting test set changes can be numerically computed at any particular point during training, using the formula from equation 5, equation 6 . This requires knowledge of the following gradients: $\frac{d\hat{y}(x, \theta(t))}{dt}$ (e.g. $\phi(x)$ for $x$ in the train and test set, as well as knowledge of the loss sensitivity $\frac{dL}{d\hat{y}(x)}$ for $x$ in the train set. Intuitively, the eNTK gives the 'influence' of training points on test points given the current parameters $\theta(t)$. These influences give the network's instantaneous changes to the output predictions on the test set, *broken down* across the test set. Note that the eNTK computation uses many of the same gradients ultimately used for the parameter update. The main advantage of computing the eNTK is that it gives breakdown of where changes to the output come from, allowing for analyses that can be used to e.g. understand the network's internal representation development. Although the eNTK computes the instantaneous influences, we find that in practice the eNTK is quite accurate for small step sizes, and can be safely used on standard networks trained with gradient descent variants.

In this article, we use the eNTK to analyze how representations in the hidden layer of the network evolve during learning of the task(s). The eNTK allows a decomposition of the instantaneous changes in the predictions of the NN over training inputs(e.g. how $\hat{y}$ can be decomposed over $x_m \in \mathcal{D}$ at any point in time). We can analyze the eNTK grouped in various ways (e.g., by input dimensions, tasks, and/or output dimensions) in order to characterize how a NN undergoing training acquires representations of the relevant information at different times over the course of learning.

## 2.2 Visualizing Learning Dynamics Using M-PHATE

We used Multislice PHATE (or M-PHATE; Gigante et al. (2019)) to visualize the evolution of representational manifolds over the course of learning. M-PHATE is a dimensionality reduction algorithm for time-series data, that extends the successful PHATE algorithm (Moon et al., 2019). PHATE captures both local and non-local structures within data, using a novel informational distance metric. Local information is captured via a custom kernel over euclidean distance, while long range information is generated via diffusion probabilities over a random walk using local kernels as the 1-step connection probabilities. This is transformed into a informational distance which measures the difference between diffusion probabilities, before being embedded into a lower dimensional space using MDS before viewing.

M-PHATE can be used to visualize internal network geometry, and that can be used to capture temporal dynamics. It does so by using longitudinal (time series) data to generate a multislice graph, and then uses PHATE to dimensionally reduce the pairwise affinity similarity kernel of the graph. By applying this to the hidden unit activations of neural networks, it can be used to visualize the representational manifolds over the

---

[2]All modern NNs perform best in the adaptive regime, and there is a significant performance gap between kernel and adaptive regime models Arora et al. (2019). The power of the adaptive regime is that it allows the NN to learn, based on the data, the basis functions that are most useful. In contrast, the kernel regime has fixed basis functions solely determined by architecture and initialization, with no dependence on the data or the task being learned.

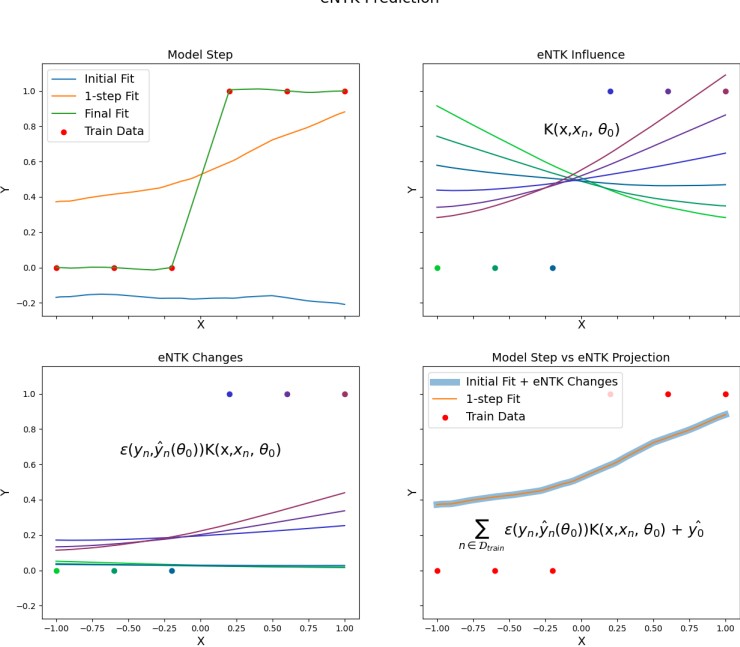

Figure 1: An explanation of a model update using the eNTK. Top Left: The model's prediction at t = 0, 1, final for $x \in [-1, 1]$. Top Right: The kernel function $K$ (t=0), or influence on the network due to each train data point $(x_n)$, tied by color (e.g. the dark purple line is the influence function $K(x, x_n)$ where $x_n$ is the dark purple dot). Bottom Left: The changes due to each influence function $K$ (t=0), which are weighted by the loss function's influence $\epsilon_n = \frac{dL}{d\hat{y}_n}$. Bottom Right: A comparison between the model's final state and the prediction made using the eNTK update. The eNTK prediction at $t = 0$ perfectly matches the true state of the network at $t = 1$. Note that while this is a one step eNTK prediction, if the network is in the kernel regime than the eNTK can predict arbitrarily far ahead.

course of training. Furthermore, by grouping analyses—for example, according to feature dimensions, tasks, or response dimensions—M-PHATE can be used to reveal how representations evolve that are sensitive to these factors.

It is important to note that our M-PHATE analysis works by analyzing the *hidden* layer activities. In contrast, the eNTK kernel computes similarities using the whole network's gradients. This means that the results of the two methods are are not directly comparable, but qualitative comparisons can be made.

## 3 Validation of eNTK Analyses of Learning in Simple Networks

We begin with a validation of the eNTK analysis by using it to predict the evolution of representations in a linear network which are tractable to standard analytic methods, the results of which can be used as a benchmark. Specifically, we replicate results from Saxe, McClelland, and Ganguli (henceforth SMG; Saxe et al. (2013)), which show that for a linear NN with a bottleneck hidden layer, the network will learn representations over that layer that correspond to singular values of the weight matrix, in sequence, each learned with a rate proportional to the magnitude of the singular value, up until the dimensionality of the bottleneck layer. Here, we show that eNTK analyses can be used to qualitatively predict this behavior from the initial conditions of the network (i.e., its initial weights and training set).

We take a simple case for illustrative purposes, using a network with an input dimensionality of 4, a bottleneck (hidden layer) dimensionality of 3, and an output dimensionality of 5. A target linear transformation $W_{target}$

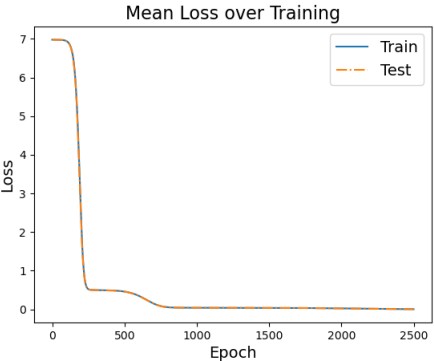

Figure 2: Loss over time in the SMG task for training and testing data (see text for network description). Note the tiered learning, with loss dropping over some regions of time while remaining near constant in others. As a simple linear task, train and test losses are nearly identical.

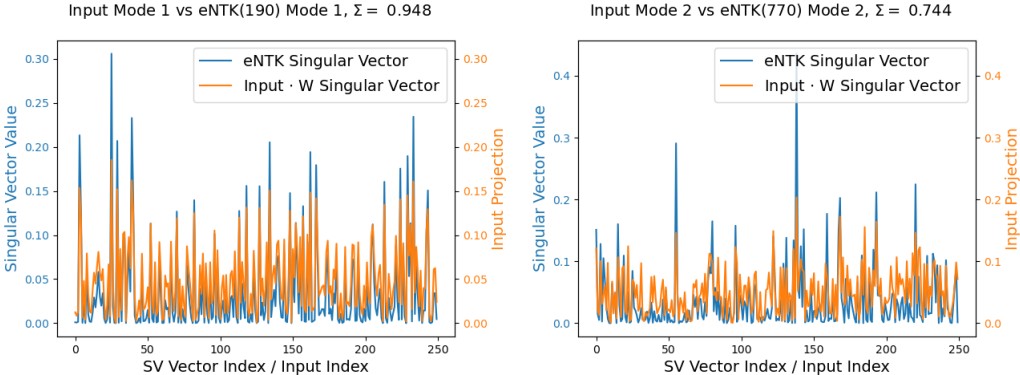

Figure 3: Comparisons between projections of training data $x_n$ onto singular vectors of $W_{target}$ (orange) and singular vectors of the eNTK (blue) at times $t = 190, 770$. At time 190 (left), the high match between eNTK and data projections onto first singular vector $W_{target}$ show that the first singular mode is being learned. At time 770 (right), the high match between eNTK and data projections onto *second* singular vector show that the second singular mode is being learned. Note the high correlations in both cases.

is generated, and then used to generate random training data $y = W_{target}x$, where $x$ is randomly drawn from white noise. The network is then trained using simple stochastic gradient descent. We call this the SMG task, and replicate their results, showing behavior qualitatively similar to that reported in their previous work (Fig 2).

What can a eNTK analysis tell us over and above the original analysis? One major benefit is that the eNTK can be used on arbitrary, rather than only on linear systems. To assess this, we numerically compare the singular vectors of the full eNTK (at specific time points $t = 190, 770$) against the projections of the data $x$ into the coordinate system spanned by the singular vectors of $W_{target}$ (e.g. ground truth for this linear system). This allows us to compare how much each data point contributes to the eNTK compared to how much it would contribute if it was perfectly learning the modes of $W_{target}$, an approach that works for linear or nonlinear systems. This comparison (Fig 3) shows that eNTK analysis gives similar results to the ground-truth linear analysis, providing support for the accuracy of the eNTK analyses. Note that the ground-truth here is only available due to this being a linear task, while the eNTK method can work for any neural network.

Finally, we conduct another analysis using the eNTK, examining how the learning of each mode changes progressively over time. The eNTK allows us to examine the patterns that are being learned at each time

point, by examining the eigenvectors of the eNTK. Consistent with the previous work (Saxe et al., 2013), the analysis confirms that eigenvectors are learned one at a time, with larger ones learned first. More interestingly, secondary learning (e.g., the mode that is being acquired at the second fastest rate, and is the second eigenvalue/vector pair of the eNTK) reveals relevant patterns, with transitions occurring at times dictated by the primary learning switching singular vectors (e.g. when the primary singular vector is learned and switches to learning the second singular vector, the secondary learning switches from learning the second singular vector to learning the third singular vector; Fig 4). These results align with changes in performance

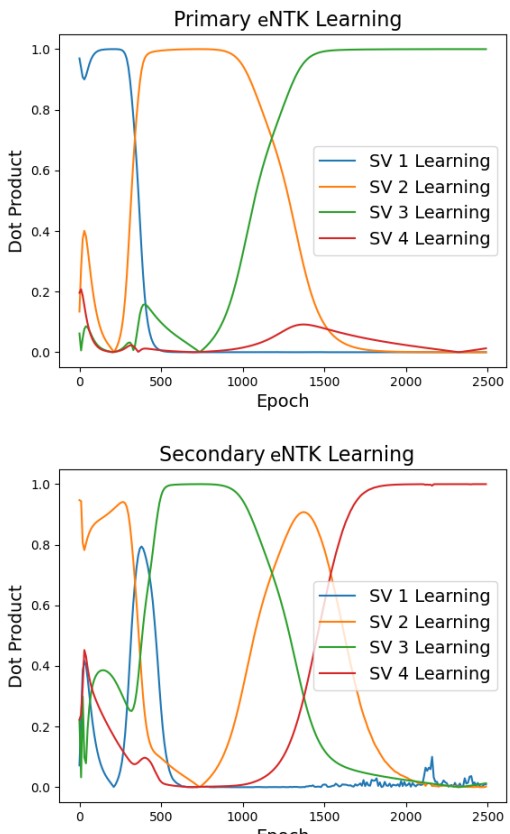

Figure 4: Match between eNTK learning (singular vector 1 - primary, shown top; singular vector 2 - secondary, shown bottom) and singular vectors of $W_{target}$. Primary learning shows a progression, where larger singular values and vectors are learned first, while the secondary learning shows a more intricate pattern with switches tied to the primary learning.

of the network. Fig 5 shows the primary learning of eigenvectors and eigenvalues over time along with model performance. This reveals that the two are closely linked, with changes in eNTK singular values *anticipating* model singular vectors aligning with the true task and accompanying decreases in task loss.

In summary, the eNTK analysis is consistent with the results reported by SMG for a linear network, using a technique that is extendable to non-linear networks. It is important to emphasize that the eNTK analysis here is *predictive*, in that the results are derived only from the conditions of the network at the time point to which the analysis is applied, but predict the representational organization of the network following subsequent learning given the training regime. Of course, the SMG theory was also predictive, but only works in the linear regime—the eNTK can readily be expanded to nonlinear applications. It also reveals interesting new patterns, particularly in the secondary learning dynamics of the network, implying that some groundwork for learning higher modes is in place before their primary learning begins.

Figure 5: Comparison between eNTK learning and model loss, showing loss vs the cosine similarity between eNTK singular vectors and $W_{target}$ singular vectors. Dotted lines show the location of $t = 190, 770$, which are the two points used for further analysis above. This shows that singular modes are learned sequentially, with the eNTK switches *preceding* and therefore *predicting* future learning, as seen via decreasing loss (e.g. Svec 2 becomes active prior to $t = 500$, while loss drops the second time after $t = 600$.

# 4 Application of eNTK Analyses to Learning and Performance in Nonlinear Networks

In the preceding section, we validated the use of eNTK analyses of learning dynamics in simple networks. Here, we explore using this technique to examine representational learning and its relationship to network performance in a more complex nonlinear network, that addresses how initial conditions influence the development of shared versus separated representations and its impact on parallel task execution (i.e., concurrent multitasking).

## 4.1 Model

### 4.1.1 Network Architecture

The network used for all further experiments is shown in Fig. 6 (cf. Musslick et al. (2020)). Notably, it has two sets of inputs, representing the stimulus and task. The task inputs are routed to both the hidden and final layer. Stimulus inputs and outputs are divided into 'pools', representing features per stimulus dimension. See Appendix A.1 for full details.

### 4.1.2 Task Environment

Each task was a one-to-one mapping from input features of a specified stimulus dimension to the response features of a specified output dimensions, mirroring response mappings in classic multitasking paradigms (e.g., Pashler 1994). Thus, the overall goal was to flexibly route inputs to outputs, with routing determined by the specific task. Multi-tasking involved multiple simultaneous routings, such that there was no input or output interference between the multiple tasks. See Appendix A.2 for full details.

### 4.1.3 Initialization and Training

In the experiments, we manipulated the initialization of all connection weights between a *standard initialization* condition (random uniform distribution in [-.1, .1]) and a *large initialization* condition (random uniform distribution in [-1, 1]). All biases were set to $b_i = -2$ to encourage learning of an attentional scheme over the task weights in which activation of a task input unit placed processing units to which it projected in the

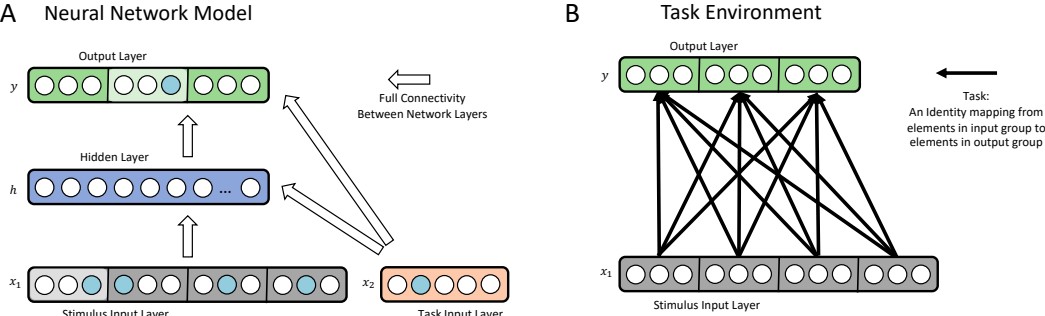

Figure 6: Network architecture and task environment. (A) Network: The input layer is partitioned into two sets, $x_1$ for stimulus inputs and $x_2$ for specification of task(s) to be performed. The stimulus set is further partitioned into four pools, each representing a stimulus dimension comprised of $m = 3$ feature units. Both input groups project to the hidden layer, comprised of $H = 200$ non-linear processing units. Both the hidden and task input units project to the output layer. The output layer is comprised of three pools of non-linear processing units, each representing a response dimension comprised of $m = 3$ response units. (B) Task: black lines show the mappings from each stimulus pool to each response pool that make up the twelve tasks on which the network was trained. Each line represents a one-to-one-mapping that the network had to learn, associating each feature within a given stimulus pool to a corresponding response in the output pool for a given task.

hidden and output layers in a more sensitive range of their nonlinear processing functions [3]. No layer-specific normalization (e.g., by batch) was used. For all experiments, networks tasks were always sampled uniformly from all available tasks. However, in each we manipulated whether training was restricted to performance of only one task at a time (*single task* condition) or required performance of multiple tasks simultaneously (*multitasking* condition), as described in the individual experiments below. In all cases, the network was trained with stochastic gradient descent (SGD) using a base learning rate of .01 for 10000 epochs, with parameters found via hyper-parameter search.[4]

We vary the initialization (between the standard and large initialization condition) in order to modify the initial inductive bias (IB) of the resulting networks. Previous results from Sahs et al. (2022) conclude that modifying the weights of later layers to be higher (as was done in our large initialization condition) will bias the resulting network to be closer to the kernel regime. There is also a rich history of work relating initialization properties to differing IBs and resulting model performance characteristics. Yang & Schoenholz (2017) was one of the earliest papers to show that varying initialization critically effected NN performance via influencing how representations are generated. More recently, the Tensor Programs series of papers, including Yang & Hu (2020); Yang et al. (2021), developed an initialization method that is *maximally* adaptive, and showed that this parametrization allows for transfer of learned hyperparameters across network scales.

### 4.2 Predicting Impact of Standard vs. Large Initialization on Representational Learning

#### 4.2.1 Effect of Initialization on Representational Learning

It has previously been observed that lower initial weights promote the formation of representational sharing among tasks that share the same input and/or output dimensions (Flesch et al., 2021; Musslick et al., 2017; 2020), consistent with other findings from work in machine learning (Sahs et al., 2022; Ding et al., 2014). Here, we sought first to validate this finding in the present architecture, and visualize it using M-PHATE,

---

[3]This exploits the nonlinearity of the activation functions to implement a form of multiplicative gating without the need for any additional specialized attentional mechanisms; see Cohen et al. (1990); Musslick et al. (2020) for relevant discussions.

[4]As the eNTK analyzes a specific network instantiation, all eNTK-based and MPHATE-based visualizations (except where otherwise noted) are based on one trial. We re-ran each experiment at least five times to confirm that there were no major qualitative changes in the results and to generate correlation metrics over multiple experiments. Experimental results are averaged over 10 trials

and then evaluate the extent to which it could be predicted by the eNTK analysis. To do so, we compared the standard initialization with the large initialization under the single task training condition.

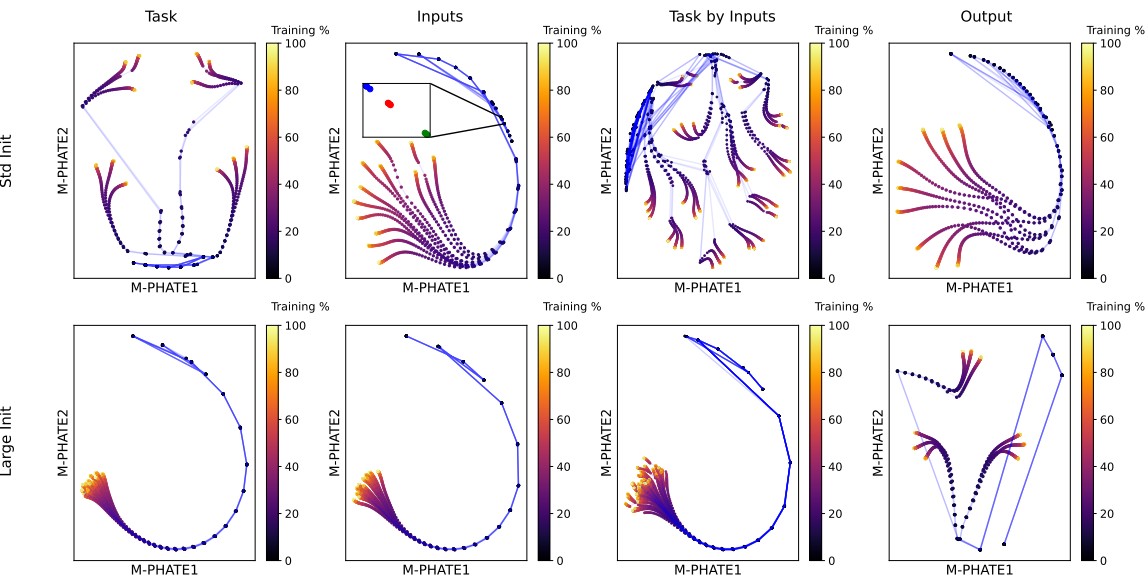

Figure 7: M-PHATE visualization of hidden unit activity in the network over course of training. Dots correspond to the relative positions of the patterns of hidden unit activities (each averaged over all input stimuli that share the same feature by which they are grouped, as explained below) at each epoch in training (darker colors represent earlier epochs in training and lighter colors later epochs; see legend to the right), with blue lines showing connections across time. Plots in the top row show standard initialization condition, and in the bottom row large initialization condition. Plots in each column show hidden unit activities grouped (averaged) along dimensions indicated by the label (e.g., in the leftmost column, labelled *Task*, hidden unit activations are grouped by task, with each dot corresponding to the average pattern of hidden unit activity over all inputs for a given task). Notice that, in all cases, in the standard initialization condition (top row) hidden unit representations exhibit highly organized structure, whereas in the large initialization (bottom row) they exhibit substantially less structure (see text for additional discussion). Inset shows M-PHATE visualization of hidden unit activity in the network with the standard initialization at t = 3 for the grouped by inputs configuration. This zoomed in view shows the transient clustering into 3 groups of 4, which collapses over time.

Fig 7 shows an M-PHATE plot of how the patterns of activity over the hidden units evolve over the course of training. The top row shows this for the standard initialization condition, with each panel showing the patterns of hidden unit activity grouped (averaged) along different dimensions. There is clear structure in the groupings, which reflects the sharing of representations for tasks that use the same inputs or outputs. For example, grouped by task (the leftmost panel), there are four clusters of three tasks each, with each cluster comprised of tasks that share the same inputs, confirmed by examining the individual elements. Similarly, grouped by input (the second panel from the left) there are three clusters of four tasks each, with each cluster corresponding to within-group position across the four pools. Notice that the grouping here gradually decoheres over time (as the clusters are transient, we show a zoomed in inset showing this phenomenon in Fig 7 (inset). Finally, grouped by output (the rightmost panel), there are three clusters of tasks that share the same output. Both effects can be seen in the *Task by Inputs* grouping (third panel from the left), in which three clusters appear early in training (dark dots), each of which is comprised of tasks that share the same output, followed later (lighter dots) by a separation into subclusters of tasks that share the same inputs. The early organization by outputs followed later with organization by inputs is consistent

with the tendency, in multilayered networks without layer-specific weight normalization, for weights closest to the output layer to experience the steepest initial gradients and therefore the earliest effects of learning.[5] These observations corroborate the general principle that lower initial weights promote representational sharing in environments for which tasks share structure.

The lower panels in Fig 7 show the results for the large initialization condition. These are in stark contrast to those for the standard initialization condition, showing little if any structure: each task develops its own representations that are roughly equidistant from the others, irrespective of shared inputs and/or outputs. The one deviation from this pattern is for grouping by output (rightmost panel), in which three clusters emerge, once again presumably reflecting the early and strong influence of the gradients on weights closest to the output of a multilayered network in the absence of any layer-specific form of weight normalization.

Together, the observations above provide strong confirmation that, in this network architecture as in many others, lower initial weights promote the development of shared representations for tasks that share structure (i.e., input or output dimensions), and showcase the utility of M-PHATE for clearly and concisely visualizing such qualitative effects. In the sections that follow, we apply eNTK analyses to show that downstream performance can be predicted quantitatively from the initial conditions.

### 4.2.2   Use of eNTK to Predict Representational Learning

First, we conduct an eNTK analysis of the standard initialization network, following initialization but prior to training (i.e., $T = 0$), to assess whether this can anticipate the effects of learning. The output of the eNTK analysis is a 500x500 matrix, corresponding to the 500 training inputs across all tasks. The M-PHATE observations shown in Fig 7 suggest that, over the course of training, the representations learned by the network's hidden units came to be clustered according to both the four input dimensions (shared across tasks) and their mapping to each of the three output dimensions (according to task). To determine whether this organization was predicted by, and can be observed in the eNTK analysis, we carried out two variants of this analysis: one in which individual training passes were sorted by the current input (aggregated over tasks), and the other sorted by task (aggregated over inputs). In both cases, the eNTK analysis is calculated per output, yielding a total of nine eNTK analyses (three units $m$ per output set $g_2$). For comparison, we also carried out the eNTK analysis without sorting. In contrast to the sorted analyses (shown in Fig 8), the unsorted analyses did not reveal any discernible structure (see Fig 16 in Appendix).

Fig 8 shows the summary results (see Fig 15 in Appendix for full results) of the eNTK analyses for the primary eigenvector (i.e., the one with the largest eigenvalue) in the standard initialization and high initialization condition at $T = 0$, sorted as described above. These reveal clustering effects in the standard initialization that correspond closely to those that emerged in the hidden units over training, as observed in the M-PHATE plots in Fig 7. Each plot shows the weighting for each training pattern on the eigenvector. When these are sorted by input features (left panels), the pattern of weightings for a given response ($m$) is the same across output pools ($g_2$) in the standard init case, consistent with the use of the same input representations across all three tasks. Complementing this, when training patterns are sorted by task (right panels), the pattern of weightings for a given response is different for each output pool in the standard init case, indicating the role of the task units in selecting which of the four input dimensions should be mapped to that output pool for each given task. Critically, this structure within the standard init is visible in the eNTK analysis *prior to training* (i.e., at $T = 0$) in the standard initialization condition but *not* the large initialization condition. Further details of other secondary analysis are provided in the Appendix.

In the standard-initialization condition, structure is visible in the eNTK analysis before any training occurs. This raises the question: to what extent is this specifically predictive of the effects that emerge during training? To address this, we analyzed the extent to which the groupings from the initial eNTK analysis predicted the structure observed in the M-PHATE hidden representations over the course of training (that is,

---

[5]To confirm this account, and rule out the possibility that early organization by outputs was because the lower dimensional structure of the outputs (three dimensions) made it easier to learn than the input structure (four dimensions), we conducted the same experiment on a network that had fewer input dimensions (three) than output dimensions (four), and observed similar results (see Appendix, Section A.7.1).

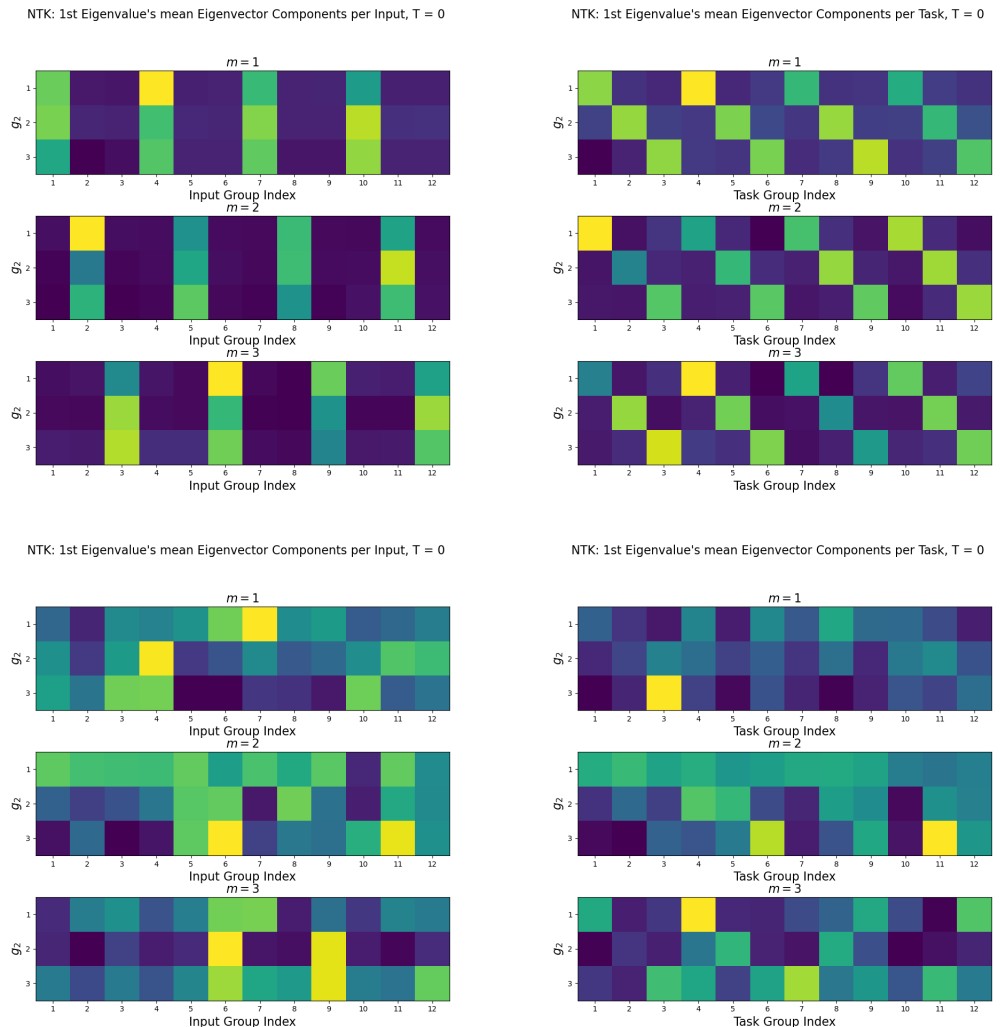

Figure 8: Mean eigenvector activations grouped by relevant eigenvector indices (x-axis), broken down by output pool ($g_2$, y-axis) and within-pool position ($m$, vertical sub-plots). Top row: standard initialization networks. Bottom row: large initialization networks. Left column: grouped by input. Right column: grouped by task. Notice the strong clustering in *both* standard init cases (see text for interpretation), but neither of the large init cases. When grouped by input (top left), the eNTK reveals that clustering is controlled by $m$ but invariant to $g_2$. On the other hand, when grouped by task (top right), the eNTK reveals that clustering is controlled by $g_2$ but invariant to $m$.

the extent to which the eNTK analysis conducted at $T = 0$, integrating only the first time step, predicted M-PHATE clustering for t>0). For the eNTK analysis grouped by input, the eNTK groups code for subgroup position within each task, as seen in Fig 8. This can easily be expressed by the M-PHATE grouping as well. For the eNTK analysis grouped by task, the grouping clearly aligns with the output pool relevant for each task, as seen in Fig 8. However, as previously discussed, the M-PHATE analysis uses the hidden layer activations, and thus cannot include output dimension information. Instead, we predict that the M-PHATE analogously uses the relevant input pool. Based on these grouping schemes, we can test the extent to which the organization predicted from the eNTK plots prior to training (at $T = 0$) predict the patterns of clustering that emerge in the M-PHATE plot at different points during training ($T>=1$). We measured the correspondence between these measures using the Adjusted Rand distance metric for between group memberships, as shown in Fig 9. The results indicate that eNTK accurately predicts M-PHATE

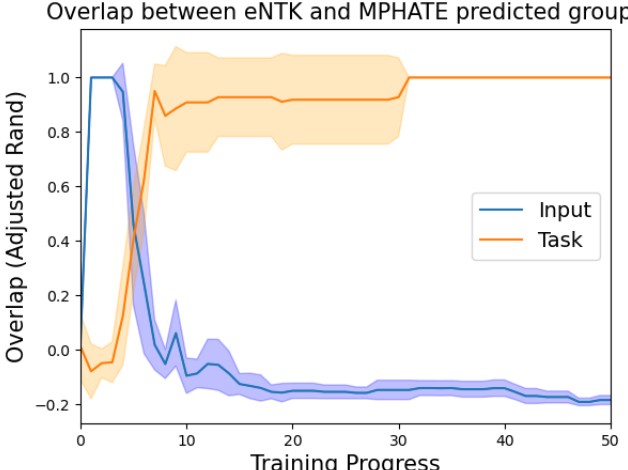

Figure 9: Overlap between groups predicted from eNTK (at $t = 0$) and groups observed from M-PHATE analysis (using both input and task-based predictions) across time (mean Adjusted Rand metric $\pm$ s.d. across trials; see text for details of analysis). The eNTK predictions for both groups are accurate, although the alignment between input-grouped and M-PHATE is transient.

structure when examined both by task and inputs. The task-grouped predictions reach a complete match in grouping by the end of training. The input-grouped predictions also align cleanly with the trajectory of representational structure in the M-PHATE analysis, reaching a prefect match in grouping early in training when structure is clearly observed in the M-PHATE analyses, followed by a diminution of the effect that parallels the dissolution of structure observed in the M-PHATE analyses.

## 5 Predicting the Effect of Initialization and Training Curriculum on Processing

The results reported above affirm the usefulness of the eNTK analyses in predicting representational learning, both in linear and non-linear networks. They also reaffirm the premise that initialization with small random weights favors representational sharing among tasks that share common structure (e.g., input and/or output dimensions). In this section, we report a further evaluation this effect, and the ability of eNTK analyses to predict not only the effects of initialization on representational learning, but also on performance. Specifically, building on previous work (Musslick et al., 2016; Musslick & Cohen, 2021), we tested the hypotheses that: i) insofar as the standard initialization condition favors representational sharing, it should be associated with faster learning of new single tasks (due to more effective generalization), but at the cost a compromised ability to acquire parallel processing capacity (i.e., poorer concurrent multitasking capability) relative to the large initialization condition; and ii) this can be predicted from the eNTK analysis at the start of training.[6] To test this, we evaluated the acquisition of multitasking performance via fine tuning of networks first trained on single task performance in each of the two initialization conditions. For each comparison, we generated two networks from the same random initialization, with the large initialization having layer 1 weights that were uniformly multiplied up by a factor of 10. We posited that although a neural network trained in the large initialization condition (and therefore biased to learn separated representations) would take longer to learn during initial single task training, it would be faster to acquire the capacity for concurrent multitasking during subsequent fine tuning on that ability, as compared to a network trained in the standard initialization condition (and hence biased to learn shared representation). Critically, we also tested the extent to which the eNTK analyses, carried out on the network *before* initial single task training, could accurately predict end

---

[6]While it is plausible that this should be the case, given that eNTK predict patterns of representational learning which only indirectly effects performance, such that it is possible that eNTK predict a different component of the variance in representational learning than is responsible for performance.

of training generalization loss and even the impact of fine tuning on concurrent multitasking performance that occurred *after* the initial training.

## 5.1 Effects of Initialization on Acquisition of Multitasking Capability

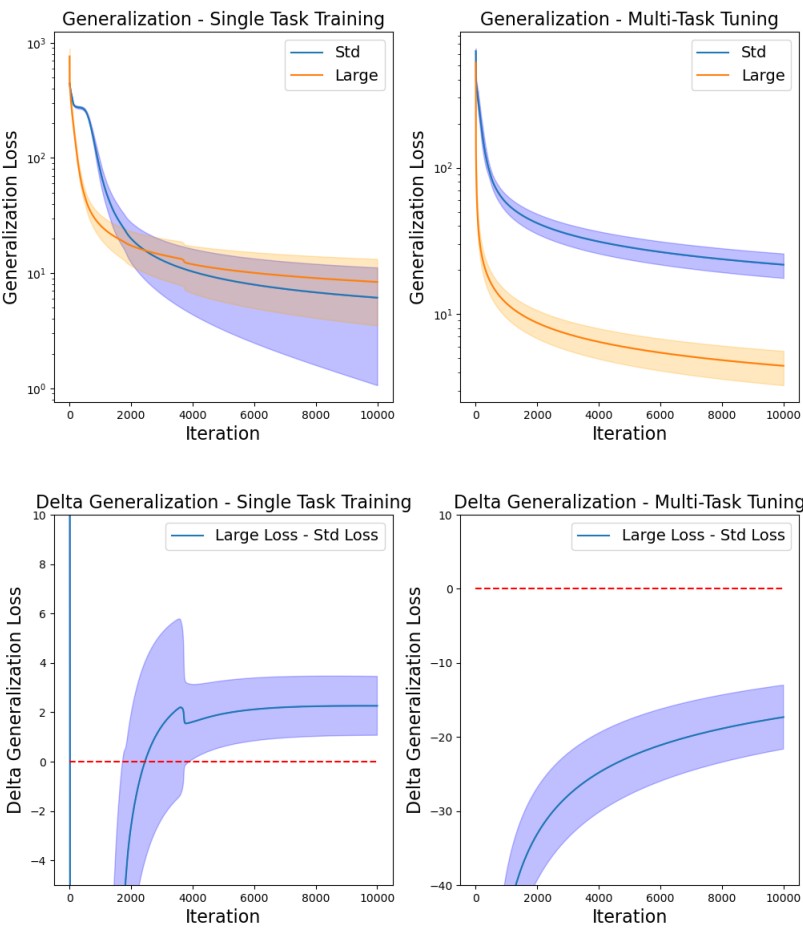

Figure 10: Top: Generalization performance (mean loss ± s.d. across trials) during single task training in the standard and large initialization conditions (left), followed by fine tuning in multitasking in each condition (right). Bottom: Difference in generalization performance (mean[difference between loss of large initialization, loss of standard initialization] ± s.d. across trials) during single task training (left), followed by fine tuning on current multitasking (right). During single task training, while the large initialization shows faster initial improvements in performance, it is quickly overtaken by the standard initialization condition. However, during subsequent fine tuning on concurrent multitasking, the large initialization condition shows a clear and sustained advantage.

To test the hypotheses outlined above, we first trained networks on single task performance in the standard and large initialization conditions. We then followed this with "fine-tuning" of each resulting network on concurrent multitasking, using the same number of training examples and epochs in each case. Specifically, we trained each network to simultaneously execute, with equal probability, either 1, 2, or 3 tasks, randomly sampled from all valid combinations (e.g. non-overlapping inputs or outputs) of the selected number of tasks.

Fig 10 shows the results for networks in the standard and large initialization conditions, both during initial single task training (left panels) and during subsequent fine tuning on concurrent multitasking performance (right panels). The upper panels show a direct comparison of the mean and standard deviation of general-

ization losses. While the basic effects are observed here, overall performance varied across different pairs of networks, as a function of the particular *pattern* of initial weights assigned to them (which was the same for each pair, and simply scaled differently for the standard and large initialization conditions). The lower panels of Fig 10 show the mean of direct comparisons between each pair of networks, that controls for differences in overall performance across the pairs. As predicted (and consistent with previous results (Flesch et al., 2021; Musslick et al., 2017; 2020), the standard initialization condition led to better generalization performance over the course of single task training, due to the development of shared representations (see Fig 7, upper panels). However, those presented an obstacle to the subsequent acquisition of concurrent multitasking performance, presumably because separated (task dedicated) representations had to now be learned *de novo*. Conversely, networks in the large initialization condition exhibited poorer overall generalization performance during single task training, due to a bias toward the learning of more separated representations (as predicted by the eNTK analyses; see Fig 7, lower panels), but it was better predisposed for the subsequent acquisition of concurrent multitasking performance, as is clearly observed in the right panel of Fig 10.

## 5.2 Predicting performance from the eNTK analysis

Next, we tested the extent to which an eNTK analysis applied to the network prior to initial single task training could predict performance during subsequent multitask tuning. To do so, we: i) created ten pairs of networks, each with a different set of weights generated for the standard and large initialization conditions (as described above; ii) applied k-means clustering with 12 groups to the multidimensional eNTK analysis of each network prior to training; and iii) quantified the clustering quality using the silhouette method (Baarsch & Celebi, 2012), with a higher value reflecting a greater degree of sharing among representations. We then trained each network using the procedure describe above, initially on single tasks, and then on multitask fine tuning. Finally, for each pair of networks, we correlated the silhouette score for the network in each condition with the difference in generalization performance between the two conditions (standard - large initialization) at the end of each training phase. If representational structure predicted by the eNTK analyses was responsible for generalization performance after each phase of training, then the correlations should be negative following single-task and positive following multitask fine tuning. This is because the standard initialization should generate higher silhouette scores (more shared representations) and correspondingly lower test loss relative to the large initialization condition after single task training (hence the negative correlation), but higher relative test loss following multitask fine tuning (hence the positive correlation); and, conversely, the large initialization should generate lower silhouette scores (more separated representations) and correspondingly higher test loss relative to the standard initialization condition after single task training (and thus, again, a negative correlation), but lower relative test loss following multitask fine tuning (again a positive correlation). The results are consistent with these predictions: the correlation was $r = -.881$ following single task training, and $r = .973$ following multitask fine tuning. This suggests that the eNTK analysis, carried out *prior to any training*, was able to reliably predict theoretically anticipated effects of initialization on generalization performance in the network observed after two distinct phases of training. This not only confirms that the eNTK analysis, carried out *prior to any training*, is able to predict the kinds of representations learned by the network in response to different initializations, but also that it can be used to predict the patterns of generalization performance associated with those representations observed in the network after two distinct phases of training. [7]

---

[7]More specifically, the analysis involved the following: Arrange the data for the 20 networks (10 each of standard and large initializations) into rows, labeled by condition (standard versus large) and with columns containing the computed silhouette score, final generalization loss following single task training, and final generalization loss following multitask fine tuning for each network. Then, add two additional columns that contained the performance of each network relative to the other member of its pair (i.e., in the other initialization condition; see Section 5.1 for the motivation for analyzing relative rather than absolute performance) after each phase of training. The values for relative performance we calculated as the difference in test loss between that condition and the other condition within the same pair (e.g., for the standard initialization: final loss of standard initialization - final loss of large initialization; for the large initialization: final loss of the large initialization - final loss of the standard initialization). Finally, compute the correlation over networks of the silhouette scores with the relative generalization scores following single task training case, and the with the relative generalization scores following multitask fine tuning.

## 6 Discussion and Conclusions

In this article, we show how a novel analysis method that examines the gradients of the network at the outset of training (determined by its initial weights and training curriculum), can be used to predict features of the representations that are subsequently learned through training, as well as the impact these have on network performance. Specifically, we show that eNTK analyses predict the extent to which a standard initialization scheme (small random weights) biases learning toward a generalizable code that groups similar inputs together (i.e., using shared representations), whereas large weight initialization predisposes the network to learn distinct representations for each task configuration (i.e., separated representations). We also confirm that, whereas the bias toward shared representations leads to improved generalization performance in the single task setting, this leads to destructive interference that impairs multitasking performance when more than one task is performed concurrently. Conversely, the large initialization scheme favors the formation of distinct task-specific representations, that facilitates the acquisition of multitasking capability.

Importantly, we show that these effects (i.e., the eventual task or input groupings) can be predicted from the eNTK eigenvectors as early as the first iteration of training, indicating that the types of representation that will be learned and their consequences on performance can be characterized before training has begun. Here, we focused on relatively simple networks, tasks, and training regimes. In general, we expect the eNTK to still be predictive, although the time window of future prediction accuracy may shrink too much to be useful in certain other applications.Evaluating the extent to which our results extend to more complex network architectures, tasks and forms of training remains an important direction for future research. We expect that this technique may be a fruitful way to advance the probing and understanding of inductive biases that influence the learning and use of representations in neural networks (Woodworth et al., 2020; Goyal & Bengio, 2022). This, in turn, may help lay the foundation for a better understanding of the respective costs and benefits of representation sharing in both biological and artificial neural network architectures.

The eNTK computation is tractable, costing only slightly more than an epoch of training. Thus, we expect the eNTK will scale to arbitrary network architectures and sizes. However, the eNTK may be prohibitively expensive for tasks where there are few, expensive training epochs (such as Large Language Models), or where there is no fixed dataset (such as RL). This expesne can be mitigated by either computing a partial eNTK (over only a portion of the train set), or a grouped eNTK (over sets of input data, such as batches). The eNTK provides a snapshot of what the network is currently learning, relating it to patterns of influence from the training data. It is less clear if these snapshots of current learning will continue to clearly describe the underlying representations as networks get larger and more complicated, with more complicated learning trajectories. It may be that multiple eNTK($t$) analyses are required as learning dynamics change in more complicated ways through time, increasing the cost and analytic complexity of the method. We plan to continue working with the eNTK, using both a more complicated network architecture, using a template based on recent work developing a more nuanced, continuous model of shared vs separated representations that takes into account the effect of extraneous 'distractor' inputs Giallanza et al. (2023).

The eNTK analysis here relied on grouping inputs by specific task or input information. However, the way our network was setup, task information is encoded within the inputs - thus, we expect that eNTK analysis will primarily be useful when combined with hypothesis of how input properties or groups influence output patterns.

The work presented here, and previous theoretical work on which it builds Feng et al. (2014); Musslick et al. (2020), suggests that sharing representations between tasks limits a network's capacity for multitasking. This has received empirical support in neuroscientific research. For example, neuroimaging studies have provided evidence that the multitasking capability of human participants is inversely related to representational overlap between tasks (Nijboer et al., 2014), and that improvements in multitasking capability are accompanied by increases in representational separation Garner & Dux (2015). The present work may aid in the development of methods that allow neuroscientists to predict the learning of shared versus separate representations before they occur. Such methods would open new paths for the quantification of individual differences or the effective evaluation of training procedures for human multitasking. Along similar lines, the

work presented here may help inform the design more effective artificial systems, by providing an efficient means of predicting the impact that initial conditions and training curriculum will have on downstream parallelization of performance.

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

# A    Appendix

## A.1    Architecture Details

It had two sets of input units: a stimulus set, $x_1$, used to represent stimulus features; and a task set, $x_2$, used to indicate which task(s) should be performed on a given trial. The stimulus set was further divided into $g_1$ pools of units, each of which was used to represent an independent stimulus dimension comprised of $m$ features along each dimension. Accordingly, each pool was comprised of $m$ units, with each feature represented as a one-hot input pattern over the group. The task set was comprised of single pool with a number of units equal to the number of tasks that could be specified (see below), and one unit used to represent each task. Both sets of input units projected to a single set of hidden units, with connection weights $w_1$ and $w_2$ for the stimulus set and task set, respectively. The H units in the hidden layer used a sigmoidal activation function $\phi$, producing an activation vector of

$$h = \phi(w_1 x_1 + w_2 x_2 + b_1)$$

where $b_1 = -2$ is a fixed bias, ensuring that the units were inhibited if no input is was provided (see Section 4.1.3). All of the hidden units, as well as the input units in the task set, projected to all of units in the output layer, with weights $v_1$ and $v_2$, respectively. The output layer, like the stimulus set of the input layer, was divided into $g_2$ pools of units, each of which was used to represent an independent response dimension comprised of $m$ responses along each dimension. Thus, each pool was comprised of $m$ units with each response represented as a one-hot input pattern over the set. Like the hidden layer, the output layer used a sigmoidal activation function $\phi$, along with output bias $b_2 = b_1$, leading to a final output of

$$y = \phi(v_1 h + v_2 x_2 + b_2)$$

or, in terms of inputs only

$$y = \phi(v_1 \phi(w_1 x_1 + w_2 x_2 + b_1) + v_2 x_2 + b_2)$$

Note that task input $x_2$ appears twice here, as there are two independent pathways involving $x_2$ (one projecting to the hidden layer and the other to the output layer).

## A.2 Task Details

A task is defined as a one-to-one mapping from the features along a single stimulus dimension to the responses of a single output dimension, reflecting response mappings in classic multitasking paradigms (e.g., Pashler 1994). This corresponded to the mapping of the $m$ input units from the specified pool $g_1$ of the stimulus group $x_1$ to the associated $m$ output unit in the specified pool $g_2$ of the output units. In aggregate, this yielded $g_1 \cdot g_2$ tasks, and thus the task set $x_2$ had that many units. For a given trial, a single feature unit was activated in each pool of the stimulus set $x_1$ (i.e., each of the $g_1$ input pools had one of its $m$ units activated). For performance of a single task, only a single unit was activated in the task set $x_2$. The network was then required to activate the output unit in the response pool corresponding to the input unit activated in the stimulus pool specified by the task unit activated in $x_2$, and to suppress activity of all other output units. For example, task 1 consisted of mapping the $m$ units of input pool $g_1 = 1$ to the $m$ units of the output pool $g_2 = 1$; as only one one of the $m$ units was active, the task consisted of mapping the active $m$th element of $g_1$ to the $m$th element of $g_2$, while outputting null responses elsewhere. For multitasking performance, two or more units in the task set were activated, and the network was required to activate the response corresponding to the input for each task specified, and suppress all other output units. Multitasking was restricted to only those tasks that shared neither an input set nor output set (see Lesnick et al. (2020) for a more detailed consideration of "legal multitasking").

We report results for a network that implemented four stimulus dimensions and three response dimensions (i.e., stimulus pools $g_1 = 4$ and output pools $g_2 = 3$).[8] This yielded a total of 12 tasks ($x_2 = 12$). Since each stimulus dimension had three features, and each response dimension had three possible responses (i.e., $m = 3$), in total the network had 24 input units ($x_1 = 12$ stimulus input, and $x_2 = 12$ task input units) and 12 output units, as well as $H = 200$ hidden units.[9]

## A.3 Path-Integrated NTK (PNTK) and ePNTK

The eNTK computes the instantaneous learning dynamics of a given test set, relative to the training inputs. What if we want to understand the entire trajectory of a dynamically learning system?

Returning to the NTK, recent work has shown that tracking the NTK along the learning trajectory/path of the model parameters $\theta(t)$ yields the Path-integrated NTK (Domingos, 2020).

Let the PNTK be denoted by $P(x, x'; t)$. Then all NNs trained in a supervised setting with gradient descent result in a (pseudo-)kernel machine[10] of the form

$$\hat{y}(x) = \sum_{m \in \mathcal{D}} P(x, x_m) + \hat{y}_0(x),$$

where the initial predictions $\hat{y}_0(x) := \hat{y}(x, t = 0)$ and $P$ is the PNTK pseudo-kernel of the form (Domingos, 2020):

$$P(x, x') := \int_0^t \epsilon_x(t') \frac{d\hat{y}(x, t')}{d\theta} \cdot \frac{d\hat{y}(x', t')}{d\theta} dt' = \int_0^t \epsilon_x(t') K(x, x', t) dt'$$

where the loss sensitivity $\epsilon_x(t)$ is now time dependent. At intermediate learning times we have

$$\hat{y}(x, t) = \hat{y}(x, t = 0) - \int_0^t dt' \sum_{m \in \mathcal{D}} \epsilon_x(t') K(x, x_m; \theta(t')),$$

---

[8]We chose a different number of stimulus and response pools to be able to distinguish partitioning of the hidden unit representations according to input versus output dimensions, or both.

[9]The number of hidden units was chosen to avoid imposing a representational bottleneck on the network, thus allowing it the opportunity to learn separate representations for the mappings of each of the twelve tasks. This was done to insure that any tendencies for the network to learn lower dimensional representation were more likely to reflect factors of interest (viz., initialization and/or training protocols) and could not be attributed to limited representational resources.

[10]The base *NTK* results in a true kernel method, as the predictions $\hat{y}$ can be written to solely depend on a weighted sum of the training data, with fixed kernel weights. In contrast, the *PNTK* is a pseudo-kernel: the path-weighted loss term brings a dependence on the test point ($x'$) to the kernel weights so that they are no longer constant, violating a requirement to be a true kernel.

Thus, the PNTK shows that the NN predictions can be expressed in terms of the NTK, weighted by the loss sensitivity function, along the entire learning path/trajectory.

The analytical PNTK is generally computationally intractable. However, the empirical PNTK (ePNTK) can be computed via discretizing the above integrals and using an approximation - we chose to use forward Euler in our experiments. Note that while the ePNTK is technically computable, it is still extremely compute and memory intensive, and we were only to compute the ePNTK for a simple test case.

Intuitively, the PNTK (and ePNTK) integral represents the summation of the predicted NTK effects over a specified time window (typically from 0 to $t$), allowing for a complete attribution of changes to the networks' test outputs to the train set.

### A.3.1 ePNTK Validation on SMG Linear System

We also validate the ePNTK on the same Saxe, McCelleland, and Ganglui (SMG) linear system used in the main text, as well as consider an alternate method.

As the SMG is a linear system, we can consider an alternate approach which exploits the fact that we are considering a simple linear system, and thus inputs can be broken down by input dimension. This allows us to compute a ePNTK value that predicts how a change in each individual input dimension affects each individual output dimension across examples, which can then be compared to the true $W_{target}$ (Fig 11). We focus on two time points: $t = 190$, that falls in the middle of the period during which the first singular vector is being acquired; and $t = 770$, that falls in the middle of the period during which the second singular vector is being acquired. The ePNTK analysis confirms that the first singular vector of $W_{target}$ is being learned at $t = 190$, the second is being learned at $t = 770$, and the first singular vector is well learned (e.g. learning acquired is significant) by $t = 190$,[11] while both are well learned (e.g. the first is also remembered) by $t = 770$.

We also extend two of our previous eNTK analysis to also include ePNTK. For the singular vector comparison, we also compare the singular vectors of the full ePNTK (again at time points $t = 190, 770$) against the projections of the data $x$ into the coordinate system spanned by the singular vectors of $W_{target}$. Fig 12 shows the ePNTK results, which show that learning progress (if the learned eigenmode is well-integrated into the network).

We also extend the eNTK analysis showing that switches in primary learning occur between loss improvements to also include the ePNTK, allowing us to analyze the relative speeds and amount of information learned. Fig 13 shows the ePNTK results, which demonstrate that the speed and amount of learning correlates with the magnitude of the singular values.

### A.3.2 Derivation from SMG

For the simple Saxe, McClelland, Ganguli (SMG) model, the ePNTK can be further broken down not only by output dimension but also by input dimension. Thus, the core ePNTK update is based off an eNTK generated by

$$NTK[i, j, k, l] = \left\langle \sum_{i,l} \left( \frac{dy_l(x_j[i])}{d\theta} \frac{dL}{dy_l(x_j)} \right), \frac{dy_l(x_k[i])}{d\theta} \right\rangle$$

where $i$ refers to input dimension, $j$ to training input index, $k$ to testing input index, and $l$ to output dimension. Thus $y_l(x)$ is the $lth$ dimension of output $y$ given input $x$, and $x[i]$ refers to using an input generated from only the $i$th input dimension of input $x$ (e.g. for $x = [1, 2, 3], x[1] = [1, 0, 0]$. This works because we have a linear system so $\sum_i f(x[i]) = f(x)$).

---

[11]What we mean by a singular vector 'is being learned' at time t is that it is the top singular vector of the eNTK(t), while 'is well learned' means that it singular value (of ePNTK(t)) is significant (compared to the maximum from $W$). This means that a well learned singular vector can also continue to be learned, while a singular vector being learned may or may not be well learned at a particular time. See Fig 13 markers for a visualization relating to this

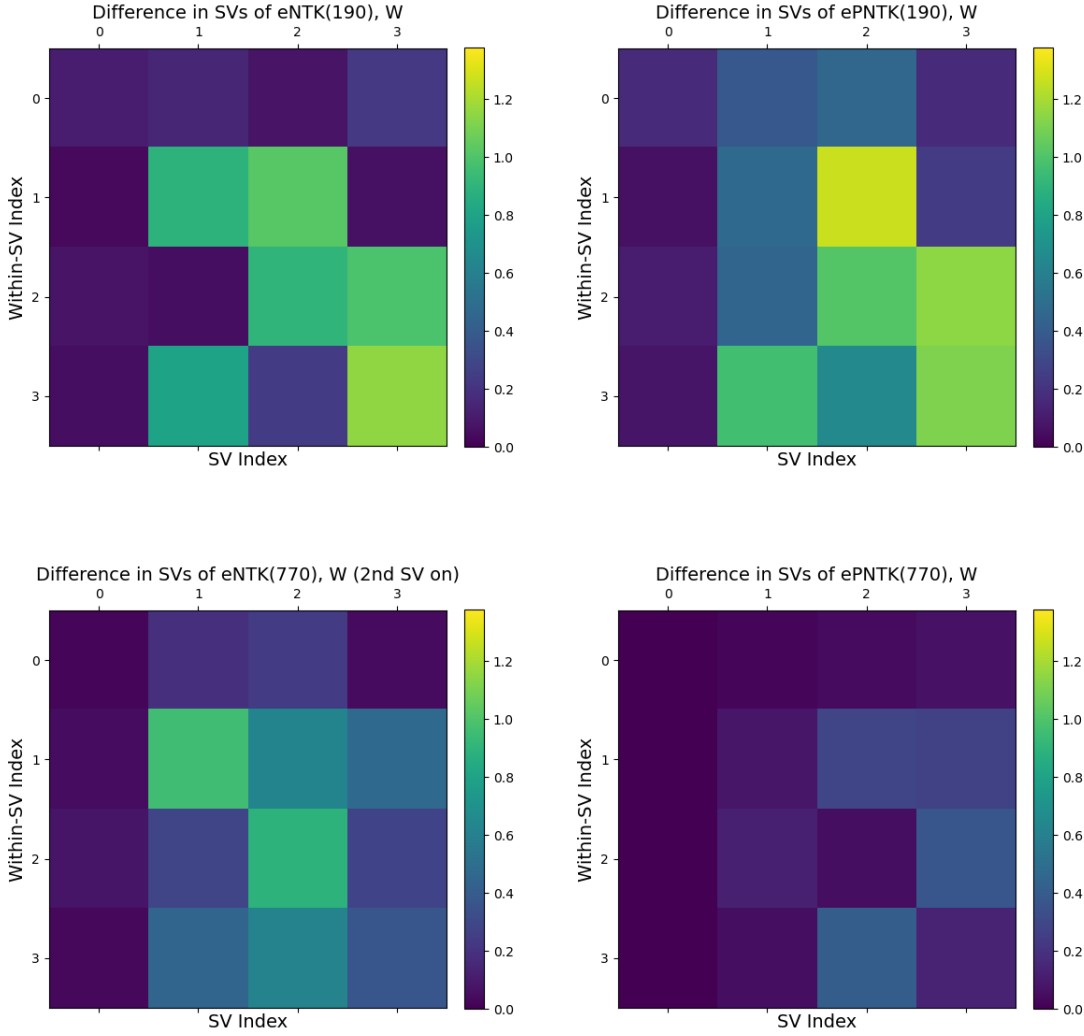

Figure 11: Differences (absolute value) between SVD singular vectors of $W_{target}$ compared to eNTK and ePNTK, at times $t = 190$ and 770. Note columns correspond to singular value index, while rows correspond to the entry within a singular value. At time 190, the ePNTK (top right) shows that the first singular vector has, for the most part, been learned, though it is still being fine-tuned, as shown by the eNTK (top left), prior to learning of the second singular vector. At time 770, the eNTK (bottom left) shows the second singular vector is being learned (notice we are comparing against the *second* singular vector of $W_{target}$), while the ePNTK (bottom right) shows that the first two singular vectors have been incorporated successfully by the model.

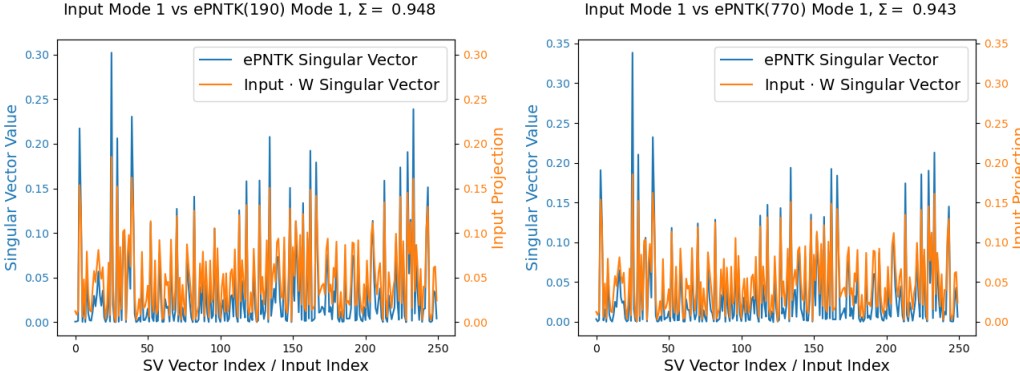

Figure 12: Comparisons between projections of $x$ into singular vectors of $W_{target}$ and singular vectors of ePNTK at times $t = 190, 770$. At time 170, the ePNTK (left) shows that the 1st mode has already been well incorporated overall. At time 770, the ePNTK (right) shows that the first mode is still remembered.

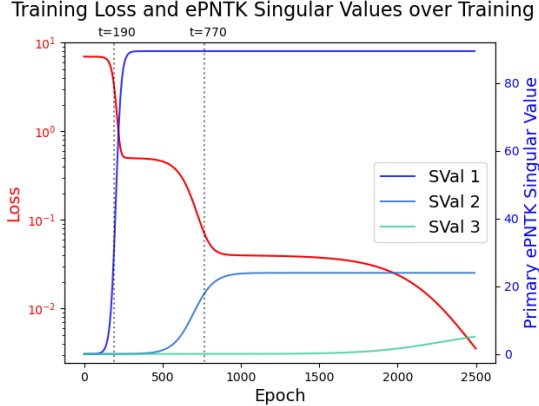

Figure 13: Comparison between ePNTK learning and model loss, showing Loss vs ePNTK singular values. Dotted lines show the location of $t = 190, 770$, which are the two points used for further analysis above. The ePNTK reveals the extent to which each mode has been learned, as seen by the jump in ePNTK singular values corresponding to the decrease in loss.

The ePNTK is accumulated from the eNTK via the update

$$PNTK- = NTK_t * \eta$$

for learning rate $\eta$. This ePNTK gives the total influence (throughout training) that dimension $i$ of training input $j$ has on the output dimension $l$ of a response to a testing example $k$. Note that summing over $i, j$ will just give the networks (learned) response to testing example $k$ across all outputs.

Since we are working in a simple linear system, we want to use the ePNTK to understand how $W_{target}$ is learned. The actual learned $W$ can be estimated from the ePNTK - the ePNTK gives the actual effect of example $j$ on future outputs, which only happens through a modification of $W$. Thus, by normalizing out by the magnitude of $x_k[i]$, and summing the response over all training $j$, we can use the ePNTK to estimate the learned change in $W$:

$$\delta W[i, l] = \sum_j PNTK[i, j, k, l]/x_k[i]$$

for any $k$. This ePNTK estimate of the learned change in $W$ should closely track the true $W_{true}$ if our ePNTK theory is correct.

Given that we are using a very simple linear system, we can simplify some of the preceding results. Our system is a linear system, given by $y(x) = xW_1W_2$, where $x$ is a vector of dimension 4, $y$ is a vector of dimension 5, $W_1$ is of dimension $4 \times 3$ and $W_2$ is of dimensions $3 \times 5$, e.g. we have a bottleneck of dimensionality 3. We generate targets according to $y = xW_{target}$, with $W_{target}$ is of dimensions $4 \times 5$, and we use MSE loss. Note that $\frac{dy_l(x[i])}{d\theta}$ is a vector of size 27: 12 parameters from $W_1$, followed by another 15 from $W_2$. Due to us splitting up and only activating only a single input and output, only a small portion of these parameters receive gradient at once - specifically, the $l$th column of $W_2$ for output activation $l$, and the $i$th row of $W_1$ for the input activation of $i$, e.g. only 6 parameters for any particular combination

$$\frac{dy_l(x[i])}{dW_1[i, :]} = x[i]W_2[:, l]$$

$$\frac{dy_l(x[i])}{dW_2[:, l]} = x[i]W_1[i, :]$$

and thus the entire set of non-zero parameters is

$$\frac{dy_l(x[i])}{d\theta} = x[i][W_2[i, :], W_1[:, l]]$$

We can then attempt to simplify the core eNTK equation. The additional complexity comes from the fact that we are summing over the indexes of $i, l$. Starting from the base equation:

$$NTK[i, j, k, l] = \left\langle \sum_{i,l} \left( \frac{dy_l(x_j[i])}{d\theta} \frac{dL}{dy_l(x_j)} \right), \frac{dy_l(x_k[i])}{d\theta} \right\rangle$$

Substiting in our previous simplification:

$$NTK[i, j, k, l] = \left\langle \sum_{i,l} \left( x_j[i] * [W_2[:, l], W_1[i, :]] * \frac{dL}{dy_l(x_j)} \right), x_k[i] * [W_2[:, l], W_1[i, :]] \right\rangle$$

Note that the notation $[W_2[:, l]W_1[i, :]]$ implies that all parameters outside of those specified are 0. Thus, we only need to worry about a subset of the parameters within the $i, l$ sum. Thus, the sum needs not be over all $i, l$ combination, but only over those such that either $i$ or $l$ matches the eNTK index. Another way of putting it is:

$$NTK[i, j, k, l] = \left\langle [\sum_i x_j[i] * \frac{dL}{dy_l(x_j)} W_2[:, l], \sum_l x_j[i] \frac{dL}{dy_l(x_j)} W_1[i, :]], x_k[i] * [W_2[:, l], W_1[i, :]] \right\rangle$$

Finally, we can convert from the eNTK to the change in $W$

$$\Delta W[i,l] = \sum_j NTK[i,j,k=0,l]/x_{k=0}[i]$$

Note $k=0$ is a generic choice - any value would work here. The idea is that $NTK[i,j,k,l]$ measures the influence of training $x_j[i]$ on output $y_l[x_k]$. However, the influence comes about through $W$, so the magnitude of the actual effect will linearly depend on the input, e.g. $x_k$. Putting the previous 2 equations together:

$$\Delta W[i,l] = \sum_j \left\langle [\sum_i x_j[i] * \frac{dL}{dy_l(x_j)} W_2[:,l], \sum_l x_j[i] \frac{dL}{dy_l(x_j)} W_1[i,:]], x_{k=0}[i] * [W_2[:,l], W_1[i,:]] \right\rangle /x_{k=0}[i]$$

we can make a few further simplifications. To start, we are using MSE error, so

$$\frac{dL}{dy_l(x_j)} = \frac{2}{ND} \epsilon_j[l]$$

where $N$ is the number of training data, $D$ is the output dimensionality, and $\epsilon_j$ is the residual (e.g. error) between the current and target $y_l(x_j)$, for a final update of:

$$\Delta W[i,l] = \sum_j \left\langle [\sum_i x_j[i] * \frac{2}{ND} \epsilon_j[l] W_2[:,l], \sum_l x_j[i] \frac{2}{ND} \epsilon_j[l] W_1[i,:]], x_{k=0}[i] * [W_2[:,l], W_1[i,:]] \right\rangle /x_{k=0}[i]$$

which we can rewrite a bit to more closely match the standard SGD formulation as:

$$\Delta W[i,l] = \sum_j \left\langle \left[\sum_i x_j[i] * W_2[:,l] * \frac{2}{ND} \epsilon_j[l], \sum_l x_j[i] W_1[i,:] * \frac{2}{ND} \epsilon_j[l] \right], [W_2[:,l], W_1[i,:]] \right\rangle$$

Looking at the true updates made by the SGD equation:

$$\Delta W[i,l] = \sum_j \frac{dL}{dy_l(x_j)} \frac{dy_l(x_j)}{dW[i,l]} = \sum_j \frac{2}{N} \epsilon_j[l] \frac{dy_l(x_j)}{dW[i,l]}$$

but we don't have $W[i,l]$ directly to update. In reality, $W = W_1 W_2$, so an update to $W[i,l]$ is dependent upon $W1[i,:]W2[:,l]$, e.g. their dot product.

$$\Delta W[i,l] = \langle \Delta W_1[i,:], \Delta W_2[:,l] \rangle + \langle \Delta W_1[i,:], W_2[:,l] \rangle + \langle W_1[i,:], \Delta W_2[:,l] \rangle$$

Assuming that our updates are fairly small and $W >> \Delta W$, we can ignore the cross term,

$$\Delta W[i,l] = \langle \Delta W_1[i,:], W_2[:,l] \rangle + \langle W_1[i,:], \Delta W_2[:,l] \rangle$$

Note that e.g.$\Delta W_1[i,:]$ will be proportional to $\frac{dL}{dy(x_j)} \frac{dy(x_j)}{dW_1[i,:]}$ - the version calculated above is for a single input index $i$ and output index $l$, whereas this one is more general. Since we have a linear system, we can get these by summation over the appropriate index (note $\Delta W_1[i,:]$ will need to be summed over $l$ but not $i$, and the reverse for $\Delta W_2[:,l]$. This leads to

$$\Delta W[i,l] = \sum_j \left( \left\langle \sum_i W_2[:,l] x_j[i] \frac{2}{ND} \epsilon_j[l], W_2[:,l] \right\rangle + \left\langle W_1[i,:], \sum_l W_1[i,:] x_j[i] \frac{2}{ND} \epsilon_j[l] \right\rangle \right)$$

which is just a reformulation of the previous equation, showing that the ePNTK exactly recovers the linear updates from SGD.

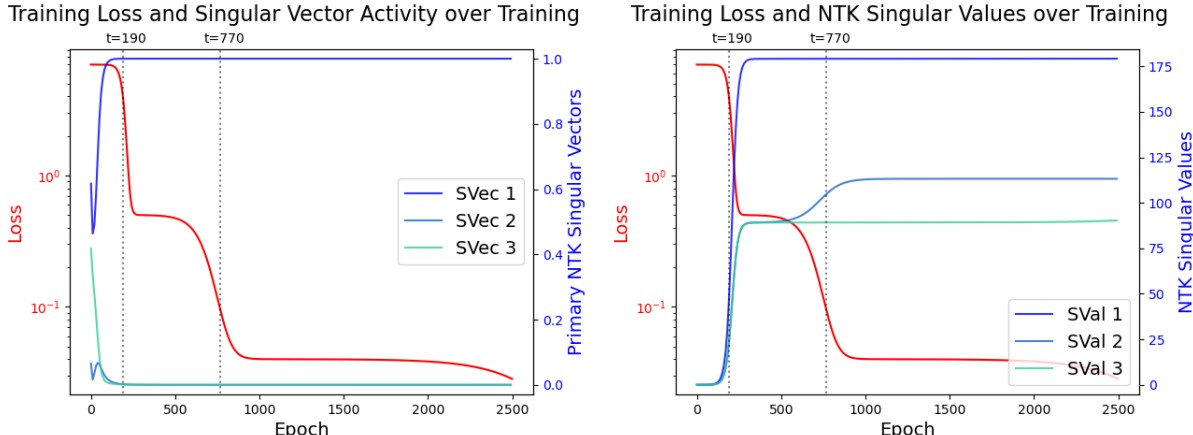

Figure 14: Figures plotting loss vs NTK SVD information. Left: Loss plotted against NTK primary singular vector overlaps with $W_{target}$ singular vectors. Note that the primary overlap is always with the first mode of $W_{target}$ across the whole learning trajectory. Right: Loss plotted against NTK top 3 singular values. Note that all 3 singular values are acquire simultaneously, and that singular value 3 reaches its maximum before the 3rd mode is actually learned (the final loss drop just before training ends).

### A.4 Differences between NTK and eNTK

Although the eNTK and NTK are closely related, they have important differences as well. The difference is due to the eNTK's inclusion of the loss sensitivity ($\epsilon$) information, which ties the eNTK to the actual learning taking place for a given set of parameters. Intuitively, the NTK kernel gives the relationship between data points, as filtered through the architecture and parameters, while the eNTK additionally modifies this by the learning dynamics.

To better illustrate this difference, we include here alternate forms of Figs 5 and 13, calculated with the NTK rather than eNTK/ePNTK (Fig 14). They reveal that the NTK is largely static over time, with primary eigenvectors unchanging, and with eigenvalues unrelated to the learning process.

### A.5 Further Experiments and Analyses

### A.6 Full eNTK Eigenvector Results

Fig 8 in the main text presents the summary statistics generated by taking the mean of the relevant eigenvector indices. The full, non-averaged eigenvectors are presented here in Fig 15, with eigenvector indices sorted as appropriate.

#### A.6.1 Unsorted eNTK

The sorting (by either task grouping or input grouping) is essential to reveal any structure within the eNTK. As a control, we here introduce the time 0 eNTK for a standard initialized network trained on single tasking (Fig 16), which shows no discernible patterns.

#### A.6.2 Full Trial Results for Standard Single Task Training

#### A.6.3 Alternative Setup - LR Tuning

The above comparison between the standard and large initializations shows a period of time during single task training where large initialization has better single task generalization performance, in contrast to our theory. This persists for a fairly long time, necessitating a long analysis (10000 iterations) in order for the standard initialization to fully converge. Part of the reason for this is that the larger weights of the large initialization

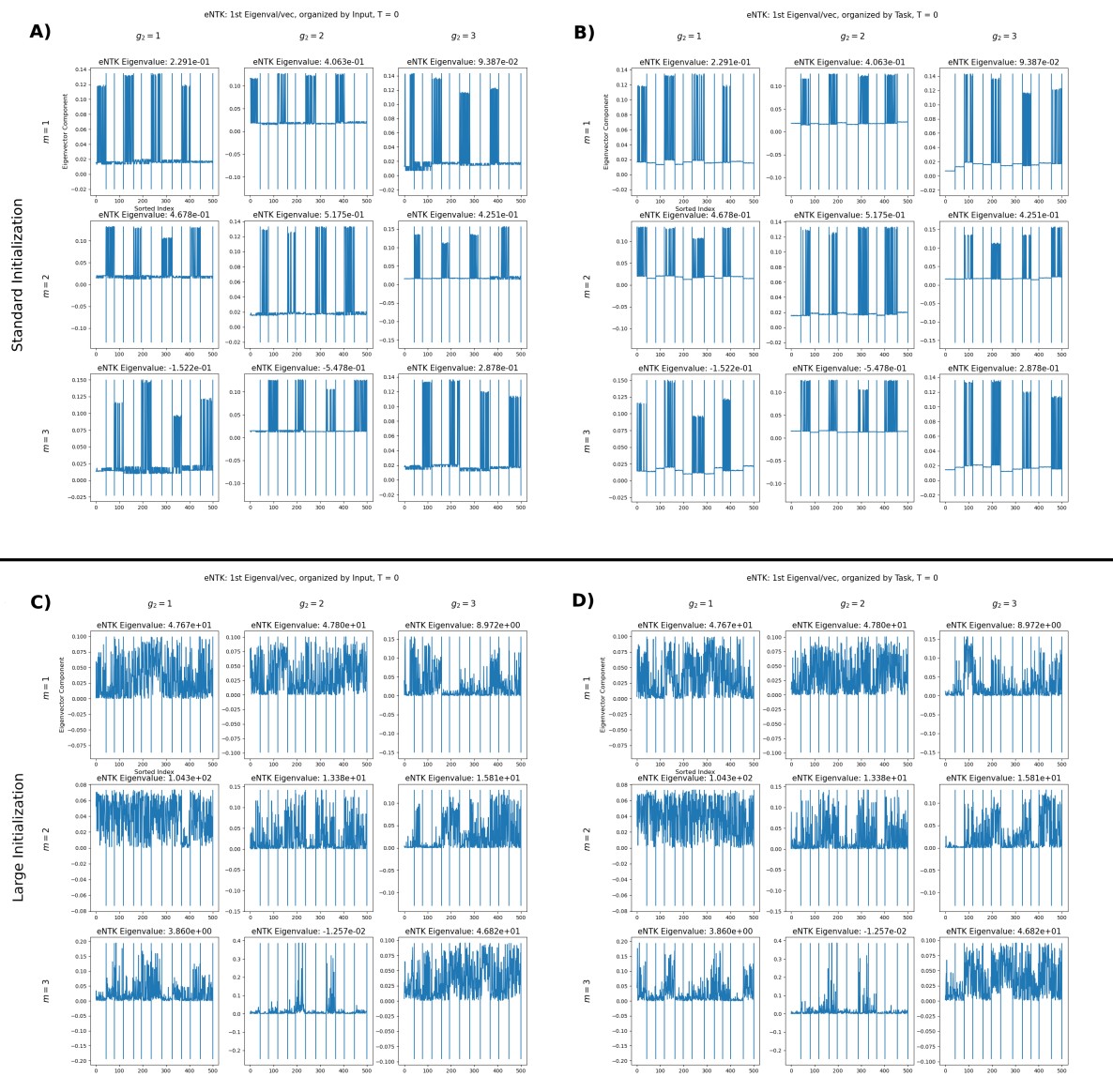

Figure 15: First eigenvector of the eNTK analysis for the output units prior to training, showing the 1st eigenvector over each training input for each output unit, with each plot's input indices (x-axis) ordered either by the relevant task (sorted by inputs; left panels) or by relevant input (sorted by tasks, right panels), across both the standard initialization (top panels) and large initialization (bottom panels). The 9 plots within each panel are organized by output pool $g_2$ and response unit $m$ within each pool. Notice the strong clustering in *both* standard init cases (see text for interpretation), but neither of the large init cases. When grouped by input (top left), the eNTK reveals that clustering is controlled by $m$ but invariant to $g_2$. On the other hand, when grouped by task (top right), the eNTK reveals that clustering is controlled by $g_2$ but invariant to $m$.

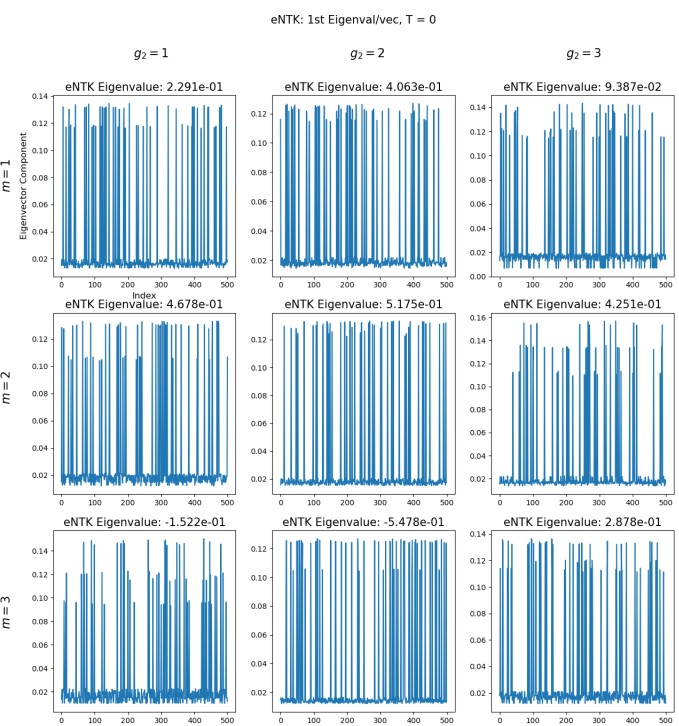

Figure 16: Ungrouped 1st eigenvectors of the eNTK at initialization, using standard initialization. Without grouping, the eigenvectors look like noise.

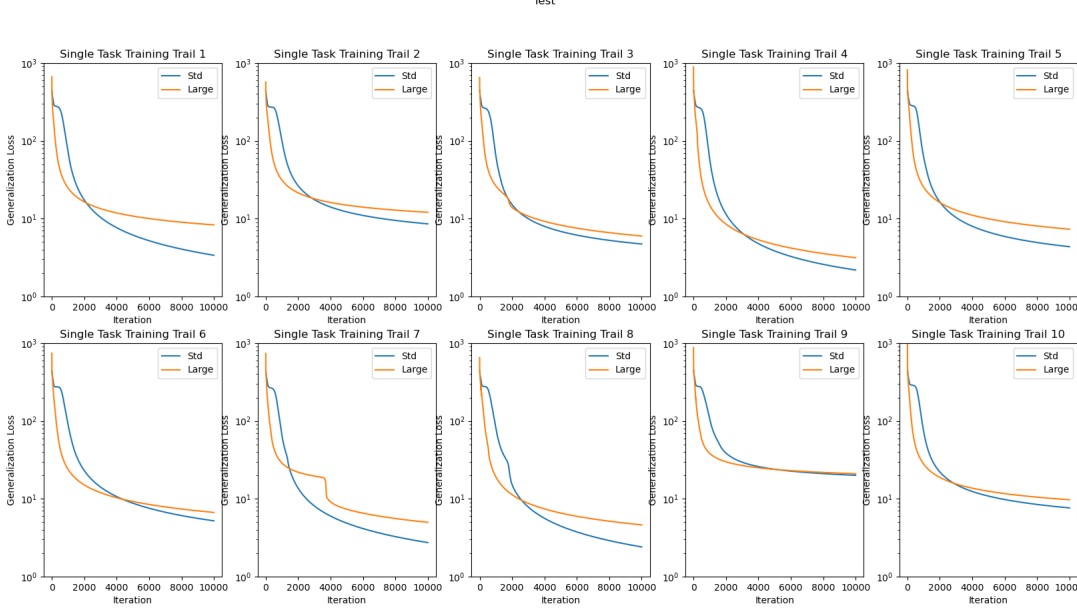

Figure 17: A plot of all 10 individually trials of the standard-setup single task training, with standardized y-axis scale. Notice that in each case, the std initialization case (blue line) eventually overtakes the large initialization case (orange line) in generalization loss performance, albeit at varying times to varying extents. However, the overall scale of the final generalization loss can vary by over an order of magnitude between trials.

condition lead to an 'effectively' higher learning rate—higher weights lead to more backpropagation of error throughout the model. We also consider an alternative approach (called the Learning Rate Tuning) where we increase the learning rate of the standard initialization in order to approximately match the starting learning rates of both initializations (learning rate set to .3 for the standard initialization case, iterations reduced to 2500 across both cases).

Fig 18 shows the results of this variant. Once again, both networks were able to learn to multitask, but the large-initialization case both learned to multitask more quickly and with improved generalization (testing loss). The improved learning rate for the standard initialization eliminated the transient area where the large initialization was better in the single tasking case, while not affecting the dominance of the large init in the multitasking case. We again focus on the per-trial differences in generalization loss in Fig 18 bottom row, which again shows a strong positive result (standard initialization better) in single task training, and a strong negative result (large initialization better) in multi-task fine-tuning.

### A.6.4 Correlation Analysis - Difference Results on LR Tuned, Combined Data

An extension of the correlation analysis, using both the LR Tuned dataset and a combined dataset (both the baseline and LR Tuned data to give a total of 20 trials).

For the LR Tuned dataset, we find a correlation of $r = -.850$ for the training case, and a correlation of $r = .952$ for the multitasking fine tuning regime.

For the combined dataset, we find a correlation of $r = -.757$ for the training case, and a correlation of $r = .932$ for the multitasking fine tuning regime.

These results are broadly consistent with the the base (non LR tuned) experimental results. In the training case, higher clustering (e.g. standard initialization) is strongly correlated with lower delta performance (e.g. superior generalization). The situation is again strongly reversed for the multitasking fine tuning case.

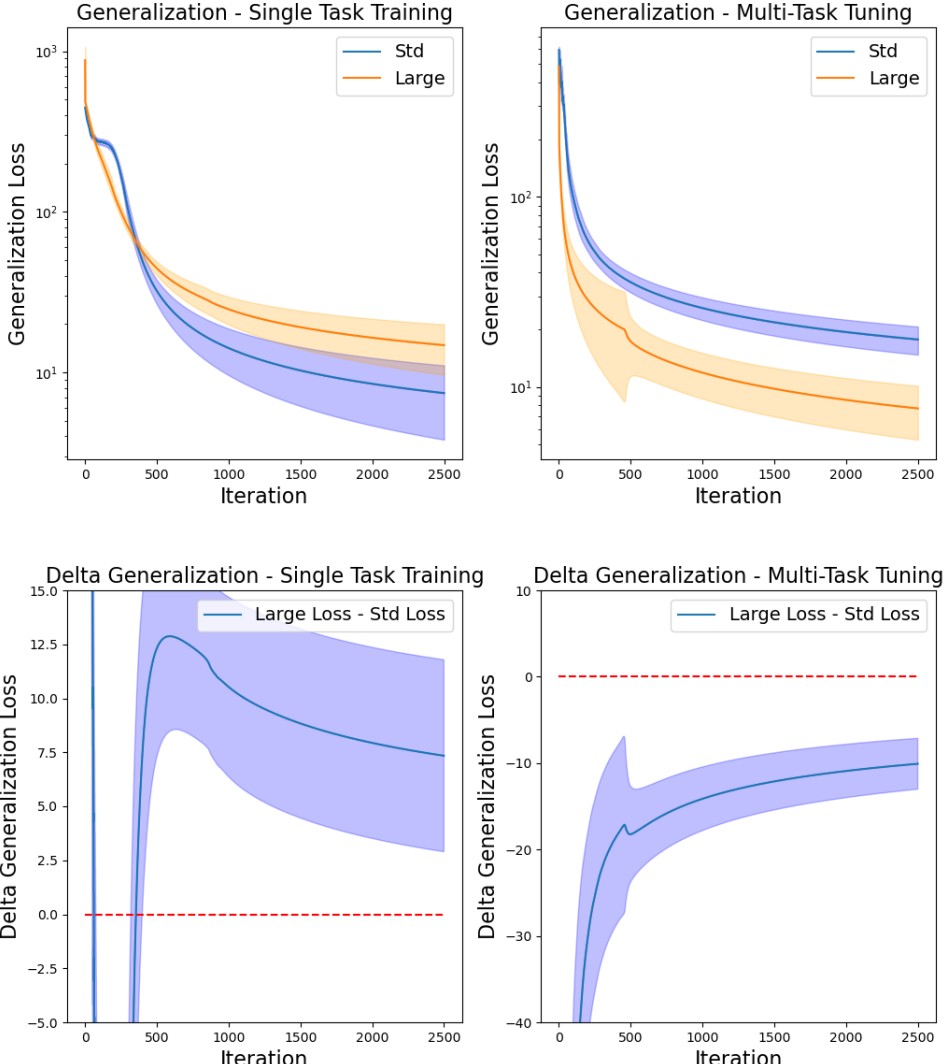

Figure 18: Generalization performance (top: mean loss ± s.d. across trials, bottom: mean difference in loss ± s.d. across trials) for the same networks and training curriculum used for Figure 10, but in which the learning rate for the standard initialization condition was increased to .03, and the training iterations reduced to 2500. This insured that the magnitudes of the initial weight updates in the two initialization conditions were approximately equal, and thus the initial speed of learning in the two conditions were also roughly comparable. The effect is that, whereas the standard initialization condition now achieves superior generalization performance earlier during training on single tasks (left panels), nevertheless the large initialization condition continues to dominate generalization performance during fine tuning on multitasking performance (right panels).

### A.6.5 Correlation Analysis - Baseline

We repeat both experimental setups a total of 10 times, then use k-means clustering with 12 groups over the multi-dimensional eNTK output, and then calculate the clustering quality using the silhouette method (Baarsch & Celebi, 2012). We then calculated the correlations between the mean silhouette score on both standard and large initialization, and the downstream fine-tuning test loss (e.g. final generalization performance). We initially report across both categories (standard and LR-tuned), before providing a per-category breakdown.

In the single task training regime, we get a get a correlation of $r = -.421$ between clustering silhouette (a measure of how well the eNTK evecs are clustered, e.g. higher for the standard init) and final generalization loss, e.g. the standard init's shared representations are correlated with generalization ability. Correlations were $r = -.224$ for the standard training setup, and $r = -.637$ for the LR-tuning training setup.

In the multitasking training regime, we get a correlation of $r = .894$ between clustering silhouette and final generalization loss, e.g. the shared init's shared representations are very strongly correlated with reduced generalization ability, as predicted. Correlations were $r = .947$ for the standard training setup, and $r = .875$ for the LR-tuning training setup.

Doing base values rather than differences dramatically reduces the power in the single task training regime, due to the aforementioned overall scale difference between different trials.

### A.6.6 Full Training Details

In Fig 10, we show the basic generalization (test loss) performance of our model across the std and large initializations for both the single-training and multi-fine tuning tasks, across two task setups (long time and learning rate tuned). Here, we additionally examine the training performance (training loss) of our models, where the dashed line corresponds to the training loss. As expected, training losses are lower than testing losses as our model can near-exactly fit seen data. In the long time case, the large initialization's larger weights allow it to learn faster even in the single task training case (top left) compared to the std initialization, although this does not correspond to increased generalization. In the learning rate matched case, the standard initialization learns better, particularly for fine tuning on multitasking (bottom left) compared to the large initialization, but again this does not lead to increase test performance. This shows that in various settings, higher training performance does not necessarily lead to better generalization ability as the network instead over-fits - the important consideration is instead the networks' inductive biases, particularly their representations.

### A.7 Effects of Initialization on Acquisition of Multitasking Capability

### A.7.1 Architectural Changes

We also briefly compared the effects of an architectural change, namely swapping from a task setup with 4 input groups and 3 output groups to one with 3 input groups and 4 output groups. In this case, there are still 12 possible tasks. In the original task, we saw that learning by inputs preceded tasks. We hypothesized that the early organization by outputs followed later with organization by inputs is consistent with the fact that in a multilayered network, absent any form of weight normalization, the weights closest to the output layer experience the steepest initial gradients and therefore the earliest effects of learning. However, it could also be due to the fact that the output dimensionality was lower, so we reversed the dimensions to confirm that the effect was unchanged, as seen in Fig 20.

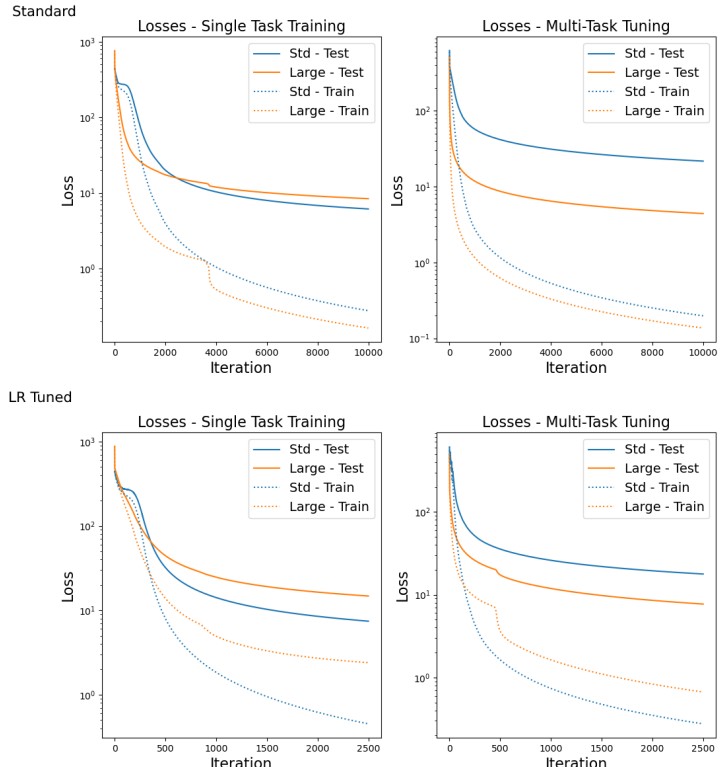

Figure 19: Top: Standard training setup. Bottom: LR-Tuning setup. The top(bottom) row corresponds to the figure in the main text(appendix), except we suppress the standard deviation and additionally show training losses as dotted lines. Training losses are unsurprisingly better than generalization losses. Interestingly, there are cases where the training loss that does better is not the generalization loss that does better e.g. in the top left the large init's effective higher learning rate allows it to quickly learn the train set, but this does not lead to improved generalization performance compared to the standard init.

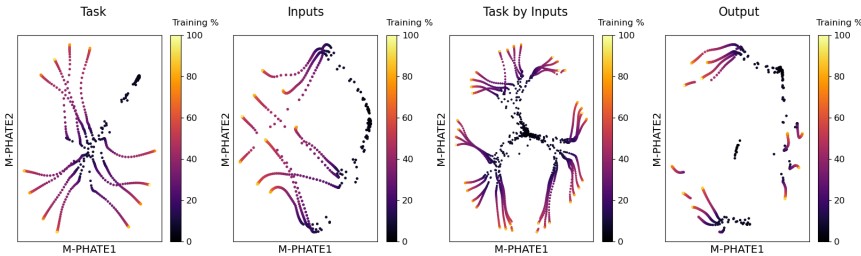

Figure 20: M-PHATE visualization of hidden unit activity in the network over course of training. Dots correspond to the relative positions of the patterns of hidden unit activities (each averaged over all input stimuli that share the same feature by which they are grouped) at each epoch in training (darker colors represent earlier points in training and lighter colors later points; see legend to the right). Plots in each column show hidden unit activities grouped (averaged) along dimensions indicated by the label (e.g., in the leftmost column, labelled *Task*, hidden unit activations are grouped by task, with each dot corresponding to the average pattern of hidden unit activity over all inputs for a given task). This figure shows an opposite input/output setup, where we have 3 input groups and 4 output groups. Similarly to the case from the main paper (4 input groups and 3 output groups), task averaging results in 4 subgroups of size 3 (compared to 4 of size 3), input averaging results in 3 groups of 3 (instead of 3 groups of 4). Task by Input grouping shows the same general pattern (learning input groups, then subdividing by task groups), as does grouping by output.

