# OpenReview forum: "A Quantitative Approach to Predicting Representational Learning and Performance in Neural Networks"
_TMLR — Rejected by TMLR_

### Review · Reviewer_TQwM · 2023-09-04

**Summary Of Contributions:**

The manuscript analyzes the effects of neural network initialization on whether the networks will learn shared representations across multiple tasks or separate per-task representations. The authors create synthetic experiments and use analyses based on Neural Tangent Kernels (NTK) and Path-Integrated Neural Tangent Kernels (PNTK) as well as a visualization-based analysis to show that standard initializations with small weights predictively lead to shared representations whereas large weight initializations lead to separated representations for different tasks. Shared representations generalize better at the end of single-task training whereas separate representations better allow later fine-tuning for multiple tasks.

**Audience:**

Yes

**Claims And Evidence:**

Yes

**Requested Changes:**

What I would find interesting is:
1) How could the method of the authors be applied to real-world scenarios, what could be predicted or tested with it, can the authors envision something here? What would be the obstacles to overcome for that?
2) Especially for readers who are not deeply informed about the different developments in deep learning theory, how does this work relate to the discussion about the appropriateness of NTK-based analyses for feature learning as mentioned e.g., in https://arxiv.org/abs/2011.14522 and https://arxiv.org/abs/2203.03466 .

**Strengths And Weaknesses:**

The manuscript comprehensively explains the author's motivation for investigating the effect of different initializations on traned neural network behavior due to  the connections of shared and separate representations to theories in cognitive science about serial, control-dependent vs. parallel automatic processing. They provide  a very detailed analysis of their synthetic experiments using complementary methods to clearly demonstrate their main claims.

A bit more discussion about how much the analyses here can show about neural network training on real-world data would be nice.
Overall, manuscript reads nicely to be, yet at times a bit longwinded, some shortening may help the overall reading.

Minor stuff:
Footnote on p.7. is cut off? text ends with "This means that"
There is an unnecessary paranthesis "four tasks each )," on page 13

---

> ### Author Response · Authors · 2023-09-23
> **Response to Reviewer 3 (TQwM)**
>
> We would like to thank the reviewer for their helpful comments, and are grateful for their clear overview of our manuscript and its aims. We address their following comments:
>
> **A bit more discussion about how much the analyses here can show about neural network training on real-world data would be nice. Overall, manuscript reads nicely to be, yet at times a bit longwinded, some shortening may help the overall reading.**
>
> We have expanded both the eNTK section and the conclusion. The eNTK section now gives additional details on the intuition, computations, and potential uses of the eNTK. The conclusion now has additional discussion about the potential uses and pitfalls of eNTK analysis on other tasks.
>
> As part of the revision, we have moved several areas to the appendix, particularly the ePNTK discussion and results, which was both the most technically complex and unnecessary towards our main goal of analyzing the shared vs separated representations task in section 4. We have also taken a pass through the introduction and conclusion sections that provide connections to and motivation from psychology literature in order to condense the most important information. However, our manuscript is still fairly long, as we feel that motivating the problem from psychology, introducing and validating the eNTK, analyzing our central shared vs separated representations task, and tying the results to wider psychology literature requires sufficient space in order to be sufficiently clear to a machine learning audience.
>
> **Minor stuff: Footnote on p.7. is cut off? text ends with "This means that" There is an unnecessary paranthesis "four tasks each )," on page 13**
>
> We have corrected these issues.
>
> **How could the method of the authors be applied to real-world scenarios, what could be predicted or tested with it, can the authors envision something here? What would be the obstacles to overcome for that?**
>
> The previous mentioned changes cover this request. In summary, the eNTK method could be applied to any network trained via gradient descent variant where gradients are computable. The eNTK decomposes the instantaneous learning of the test set to influences from the training set (or the sub-section of it analyzed by the eNTK method). The eNTK is based on the instantaneous gradients, so as step sizes increase the difference between idealized and actual network dynamics changes. In addition, on more complicated networks the instantaneous learning may be less relevant to overall learning dynamics, if instantaneous learning is highly variable over time. In this case, multiple eNTK analyses at different time points may be required, potentially with a more sophisticated analyses method which accounts for having multiple eNTK snapshots.
>
> **Especially for readers who are not deeply informed about the different developments in deep learning theory, how does this work relate to the discussion about the appropriateness of NTK-based analyses for feature learning as mentioned e.g., in https://arxiv.org/abs/2011.14522 and https://arxiv.org/abs/2203.03466 .**
>
> We have clarified our technique, renaming it to the empirical NTK (eNTK) to further differentiate it from previous, theoretical NTK results which primarily rely on taking an infinite width limit. We thank the reviewer for bringing these relevant papers to our attention, and have added references to them in the section motivating our changing of initialization in order to modify the inductive bias.
>
> As an aside, the papers show an initialization that brings maximal feature learning, (e.g. maximally adaptive), as opposed to the frozen feature (kernel) regime that the original NTK theory was derived for. The eNTK would still be effective at computing instantaneous learning kernels in this maximally adaptive regime, but these instantaneous snapshots might be less valuable to the extent that the maximal feature learning leads to variation in what is learned over time. Modern machine learning tends towards the kernel regime as network sizes increase and as learning time proceeds, allowing the eNTK to be a useful analytical technique - but this might change if the maximally adaptive regime became the new standard. This is an interesting future research direction.

---

> > ### Comment · Reviewer_TQwM · 2023-10-23
> >
> > Thanks for your answers and revisions.
> > Upon reading the other reviews I  think two points seemed concerning to me:
> > 1) Is empirical NTK claimed to be novel? It read like this to me but seems questioned by the other reviewer?
> > -> Eigenvector analysis of the empirical neural tangent kernel has been conducted previously in the literature, most notably by Loo et al. (2020) who provide quite striking visualizations of the empirical ntk in convolutional networks.
> >
> > 2) Predictiveness/Fig5
> > I am also not entirely sure about predictiveness claim here, like the dotted line seems to be always after the start of a loss decrease? not directly preceding it? Maybe could be made visually more clear what exactly is claimed to be predicted.
> >
> > There is also typo in Fig3 caption, guess should be time 190 : "at times t = 190, 770. At time 170 (left)"
> >
> > You noted the TMLR statement about journal scope  "TMLR emphasizes technical correctness over subjective significance, to ensure that we facilitate scientific discourse on topics that may not yet be accepted in mainstream venues but may be important in the future…’" I agree this needs to be taken into consideration, this also makes the two questions above to me even more important as they relate to precision of the claims.

---

> > > ### Author Response · Authors · 2023-11-08
> > > **Final Response to Reviewer 3 (TQwM)**
> > >
> > > We would like to thank the reviewer for their additional comments, and for pointing out a typo. We would like to address our responses to the following main concerns:
> > >
> > > **Novelty of the eNTK**
> > >
> > > The reviewer raised concerns from another reviewer that the eNTK was not novel. Although the eNTK is closely related to the NTK, they do measure different properties of the network. Previous work (such as the mentioned Loo et al. (2022)) carried out computational analysis of the finite-width NTK, not the eNTK (although Loo et al. did do an NTK eigenvector analysis, similar to our eNTK SVD analysis).
> > >
> > >  We have made further revisions to the eNTK section in order to make the difference more clear - this includes fixing a mistake that may have resulted in some of the confusion. In addition we have added a new section in the appendix that directly compares and contrasts an eNTK analysis to an analysis done using a computational evaluation of the NTK, while providing additional context and intuition to the difference between the eNTK and NTK.
> > >
> > > **Predictiveness of eNTK (Fig 5)**
> > >
> > > For the eNTK predictiveness in Fig 5, we are referencing the fact that the active singular vector (the blue lines) changes before the loss begins to drop - the dotted lines are just references to t = 190,770 which were used in Fig 3 (in fact, these were chosen to be in the middle of the loss drop e.g. during actual learning). For example, singular vector 2 becomes the active singular vector before t = 500, while the second loss drop does not begin until around t = 600.
> > >
> > > We have updated the caption to make this clearer.

---

### Review · Reviewer_P5dv · 2023-09-07

**Summary Of Contributions:**

This paper studies the learning dynamics of a set of feedforward multilayer perceptron architectures, and identifies conditions under which properties of the features of a randomly initialized network from this architecture class can predict input clustering in the learned representation at the end of training. It additionally studies the trade-offs between fast adaptation and effective concurrent multitasking in settings where the network must learn to achieve multiple tasks. The primary analysis techniques used by the paper include spectral analysis of the (empirical) neural tangent kernel and a dimensionality reduction technique to visualize the trajectories of the network's learned features.

**Audience:**

Yes

**Claims And Evidence:**

No

**Requested Changes:**

### Weaknesses

The paper has three main weaknesses: (lack of) clarity of the text and figures, limited evaluation to extremely small and artificial network architectures, and failure to connect some of these findings into the existing literature on neural network training dynamics. I list a few

  - The paper frequently refers to "NTK analysis", which I at first interpreted to refer to the infinite-width limit and corresponding initialization scheme that gives a fixed kernel throughout training. However, based on the last equation on page 4, where the kernel is given as a function of time, it appears that the paper refers to the *empirical* NTK, i.e. the dot product between gradients of samples at time t. Please clarify this in the text.
   - Figure 1 is missing a legend describing what the colours of the top right and bottom left figures indicate. It is also not clear what data point x is referring to in the in-axis equations.
   - In the top equation on p6, it is not clear how the dot product between two vectors that depend on time t' became a kernel that depends on time t. The notation y'(x') is also confusing -- I am assuming y' is not a derivative but rather just meant to indicate the label of x'?
  - The integral in the PNTK  appears to be highly nontrivial to solve, but no discussion is provided to clarify how it is approximated. Do you sample the NTK over a training trajectory and take an average? It also seems a bit strange that the NTK is defined using the gradients with respect to the outputs, while the PNTK is defined with respect to the derivative of the loss. It's possible that this is deliberate, but if so it would be useful to mention this more explicitly.
  - The NTK analysis is proposed in the context of an infinitesimal learning rate, but it is not clear whether this is exposition or whether the network trained in Section 3 is indeed trained via a differential equation solver or whether it is trained with standard finite-step-size gradient methods.
  - Either the train and test losses are identical (in which case I recommend using '-.' linestyle for test loss), or Figure 2 is missing the train loss
  - The labels of Figure 3 aren't very clear, and while it seems as though some notion of distance between the singular values of the NTK and the matrix W is decreasing, I can't gather from the paper text or the figure description what this notion of distance is.
   - Figure 4 is also quite difficult to interpret -- three of the figures appear identical, and the axis label makes it difficult to understand whether the singular vector value refers to the value of a projection onto the singular vector or the singular value associated with this singular vector (in which case I would have expected the values to increase/decrease monotonically with the index.)
   - The neural architectures studied in both Section 3 and Section 4 are very small and shallow. It isn't clear to me whether the findings from these sections will also hold for larger models.
   - I am aware of a number of similar analyses (in particular that of Saxe et al.) in the literature which resemble the NTK analysis conducted in Sections 3 and 4, but I wasn't able to find much discussion of these in the related work or background sections of the paper. It would be useful to understand how the analysis conducted in this paper differentiates itself from what is already out there.
   - The partitioning of the input space into task-specific dimensions is very idealized, and I would like to see analyses of more realistic multi-task settings before having confidence in the generality of the claims put forward concerning multitask learning.
    - The analysis of the importance of initialization scale bears resemblances to a number of prior works studying the importance of initialization on feature geometry and learning dynamics in neural networks, for example a number of papers by Greg Yang study the implications of weight initializations on feature learning, c.f. "Mean Field Residual Networks: On the Edge of Chaos" and "Feature learning in infinite-width neural networks". Can the authors clarify how their findings compare to prior work?
   - The paper quite long (21 pages vs the standard 10-12 in the TMLR formatting) but I don't think the content requires this. It would benefit from a review of the figure placement and more concise text.

### Requests for improvement:

The authors should address the concerns listed above. At a high level, I am looking for the following:

  1) Improved clarity, particularly regarding what is being measured in the figures and how the networks are being trained. At the moment I don't know how to interpret most of the figures illustrating dot products with eigenvectors, which makes it difficult to understand the significance of these results and the utility of the proposed analytic tools.

  2) Placement in the context of the broader literature on neural network training dynamics. For example, I don't understand what novelty is being provided by Section 3 beyond the results given by Saxe et al. The discussion of initialization scale would also benefit from better grounding in the literature.

  3) If the goal of the paper is to present spectral analysis of the empirical NTK as a useful tool for the study of neural networks, it is critical to show that this can provide interpretable insights in models that at least resemble those used by the broader community. Since the paper doesn't provide any particularly surprising or novel empirical observations, in my view the value of the paper would come from showing that spectral analysis of the empirical NTK can tell us something meaningful about what the network is learning at a given point in time, and what it will learn in the future. But for this analysis to be useful, I would need to see that it can be applied to non-trivial architectures.

  4) The study of the multi-task setting is where the paper best distinguishes itself from related work, and in my view this setting would be worth expanding, while prior sections can be condensed.

**Strengths And Weaknesses:**

- The paper contains a few very nice visualizations, in particular:
      - The M-PHATE visualizations are quite striking.
      - Figure seven clearly shows the problem setting in a clean and easy-to-interpret way.
  - The expository connection to neuroscience is intriguing and raises an interesting concern regarding multitasking that I think has not arisen much in the machine learning literature yet as we typically only ask our networks to output the answer to one question at a time. However, especially with the advent of large language models which may cache intermediate outputs so as to accelerate inference, this might become more relevant in the near future.
   - I think that studying the empirical NTK throughout training is an insightful strategy to understand what a neural network is learning at any given time, and Figures 5 and 6 do a very nice job of illustrating this.
   - The findings showing how the NTK/PNTK can predict later feature clustering under certain initialization schemes is quite interesting and reminds me of phenomena related to large-learning-rate early training properties such as linear mode connectivity and the catapault mechanism. It would be interesting to see if one can explicitly characterize the boundary at which the NTK ceases to predict final trained features in this setting, and try to generalize these findings to arbitrary network architectures.

---

> ### Author Response · Authors · 2023-09-23
> **Response to Reviewer 2 (P5dv) (1/3)**
>
> We would like to thank the reviewer for their helpful comments, particularly with regard to clarity and missing links to existing literature. We addressed their following comments and requests:
>
> **The paper frequently refers to "NTK analysis", which I at first interpreted to refer to the infinite-width limit and corresponding initialization scheme that gives a fixed kernel throughout training. However, based on the last equation on page 4, where the kernel is given as a function of time, it appears that the paper refers to the empirical NTK, i.e. the dot product between gradients of samples at time t. Please clarify this in the text.**
>
> We rename our NTK technique as the empirical NTK (eNTK) based on the reviewer’s suggestion. We clarify in the text that the eNTK differs from previous theoretical NTK results. We have also updated the entire eNTK section to be easier to follow.
>
> **Figure 1 is missing a legend describing what the colours of the top right and bottom left figures indicate. It is also not clear what data point x is referring to in the in-axis equations.**
>
> We have reworked the caption to make the colors meaning clearer, as well as to differentiate the visualization of the line x from the training points x_n
>
> **In the top equation on p6, it is not clear how the dot product between two vectors that depend on time t' became a kernel that depends on time t. The notation y'(x') is also confusing -- I am assuming y' is not a derivative but rather just meant to indicate the label of x'?**
>
> The notation of the eNTK and ePNTK has been standardized to better match each other and be clearer. The ePNTK is also less relevant to our main task and results, and has been moved to the appendix.
>
> **The integral in the PNTK appears to be highly nontrivial to solve, but no discussion is provided to clarify how it is approximated. Do you sample the NTK over a training trajectory and take an average? It also seems a bit strange that the NTK is defined using the gradients with respect to the outputs, while the PNTK is defined with respect to the derivative of the loss. It's possible that this is deliberate, but if so it would be useful to mention this more explicitly.**
>
> The appendix ePNTK section has clarifying text added that the ePNTK is also approximated empirically. We also note its high computational cost, which is why it was only solved for the small exploratory test case and not used in the main shared vs separated representations task
>
> **The NTK analysis is proposed in the context of an infinitesimal learning rate, but it is not clear whether this is exposition or whether the network trained in Section 3 is indeed trained via a differential equation solver or whether it is trained with standard finite-step-size gradient methods.**
>
> As part of the expanded paragraph explaining how the eNTK is computed and used in practice, we note that the eNTK computes the instantaneous rates, but that discretization error is quite small for standard step sizes.
>
> **Either the train and test losses are identical (in which case I recommend using '-.' linestyle for test loss), or Figure 2 is missing the train loss**
>
> This figure has been updated as recommended.
>
> **The labels of Figure 3 aren't very clear, and while it seems as though some notion of distance between the singular values of the NTK and the matrix W is decreasing, I can't gather from the paper text or the figure description what this notion of distance is.**
>
> Figure 3 has been moved to the appendix. It was an alternative, more complicated method to validate that the eNTK was accurate. Instead we now rely primarily on figure 4 (using the old figure numbering). The caption and axis labels have also been updated to make the comparison more explicit.
>
> **Figure 4 is also quite difficult to interpret -- three of the figures appear identical, and the axis label makes it difficult to understand whether the singular vector value refers to the value of a projection onto the singular vector or the singular value associated with this singular vector (in which case I would have expected the values to increase/decrease monotonically with the index.)**
>
> This figure has also been updated - their are now two y axis labels, making it clear that we are measuring eNTK singular vector  components (blue, left y axis) against inputs projected onto singular vector of W_target (orange, right y axis), while the figure caption, legend, and labels have also been reworked to make the comparison clearer.

---

> > ### Author Response · Authors · 2023-09-23
> > **Response to Reviewer 2 (P5dv) (2/3)**
> >
> > **The neural architectures studied in both Section 3 and Section 4 are very small and shallow. It isn't clear to me whether the findings from these sections will also hold for larger models.**
> >
> > The architecture from Section 3 is intentionally simple, to show how the eNTK analysis works and can replicate findings from a previous work. The architecture in section 4 is the minimal complexity needed to capture the relevant shared vs separated representations, inspired by findings from psychology literature, bringing the ideas of concurrent multitasking to a machine learning context. A new section in the conclusion discusses the potential benefits and pitfalls of applying the eNTK to larger, more complicated networks.
> >
> > **I am aware of a number of similar analyses (in particular that of Saxe et al.) in the literature which resemble the NTK analysis conducted in Sections 3 and 4, but I wasn't able to find much discussion of these in the related work or background sections of the paper. It would be useful to understand how the analysis conducted in this paper differentiates itself from what is already out there.**
> >
> > Section 3 is a replication of previous results from Saxe et al; the goal of which is to show how the eNTK works, finding similar results to a previously studied network, with the caveat that the eNTK is applicable to generic networks, while the techniques from the replicated paper required a linear setting. The main analysis is then done on the nonlinear shared vs separated representations task.  We have made sure to delineate the eNTK from previous theoretical NTK results, as well as added additional references to related works.
> >
> > **The partitioning of the input space into task-specific dimensions is very idealized, and I would like to see analyses of more realistic multi-task settings before having confidence in the generality of the claims put forward concerning multitask learning. - The analysis of the importance of initialization scale bears resemblances to a number of prior works studying the importance of initialization on feature geometry and learning dynamics in neural networks, for example a number of papers by Greg Yang study the implications of weight initializations on feature learning, c.f. "Mean Field Residual Networks: On the Edge of Chaos" and "Feature learning in infinite-width neural networks". Can the authors clarify how their findings compare to prior work?**
> >
> > The shared vs separated representations task of Section 4 was designed as a low complexity example that incorporated the main ideas from psychology, namely that concurrent multi-tasking can induce additional computational burden, and the way in which shared vs separated input representations interact with this. We agree that this is a highly simplified setting, and have expanded our conclusion to discuss ways in which  a similar analysis in a more realistic psychology-inspired network can be carried out in future work, where the difference between shared vs separated representations is more continuous rather than binary.
> >
> > We thank the reviewer for their relevant references, and have incorporated them into the section where we discuss the initialization scale difference and how it is expected to change the resulting networks’ inductive biases.
> >
> > **The paper quite long (21 pages vs the standard 10-12 in the TMLR formatting) but I don't think the content requires this. It would benefit from a review of the figure placement and more concise text.**
> >
> > We realize that our paper is quite long, as we are attempting to bring ideas from psychology to machine learning while simultaneously introducing and validating the eNTK, all of which requires significant exposition to promote clarity. Various changes made (moving content to the appendix, reworking or simplifying figures, and a pass to remove redundancies) have all led the revision to be somewhat shorter than the original.
> >
> > **Improved clarity, particularly regarding what is being measured in the figures and how the networks are being trained. At the moment I don't know how to interpret most of the figures illustrating dot products with eigenvectors, which makes it difficult to understand the significance of these results and the utility of the proposed analytic tools.**
> >
> > Figure captions have been overhauled to better explain what is being shown, and include the key takeaway for each figure. Additionally, we have made a pass through the experiments section and the eNTK section to make it clear that both the eNTK analyses and network training use gradient descent, not gradient flow.

---

> > > ### Author Response · Authors · 2023-09-23
> > > **Response to Reviewer 2 (P5dv) (3/3)**
> > >
> > > **Placement in the context of the broader literature on neural network training dynamics. For example, I don't understand what novelty is being provided by Section 3 beyond the results given by Saxe et al. The discussion of initialization scale would also benefit from better grounding in the literature.**
> > >
> > > Section 3 is a replication of results from Saxe et al using the eNTK, in order to show how the eNTK works and what results it can generate, most notably the fact that eNTK analysis is predictive of future learning. We have also added in several new references supporting our usage of initialization scale to change inductive bias.
> > >
> > > **If the goal of the paper is to present spectral analysis of the empirical NTK as a useful tool for the study of neural networks, it is critical to show that this can provide interpretable insights in models that at least resemble those used by the broader community. Since the paper doesn't provide any particularly surprising or novel empirical observations, in my view the value of the paper would come from showing that spectral analysis of the empirical NTK can tell us something meaningful about what the network is learning at a given point in time, and what it will learn in the future. But for this analysis to be useful, I would need to see that it can be applied to non-trivial architectures.**
> > >
> > > The primary goal of this paper is to use various techniques, including M-PHATE and eNTK, to validate our hypothesis on the shared vs separated representations task. More broadly, we desire to bring notions of concurrent multi-tasking, control, cognitive flexibility, and how these interact with input representations, from the psychology community to the machine learning community. Thus, we do not focus on applying techniques to state of the art machine learning models. We believe this is appropriate for TMLR, as TMLR’s overview states: ‘TMLR emphasizes technical correctness over subjective significance, to ensure that we facilitate scientific discourse on topics that may not yet be accepted in mainstream venues but may be important in the future…’. We understand that the eNTK method may be of more general interest, and have added new content in its section and the conclusion expanding upon intuition, uses, and potential pitfalls of expansion to larger or more complicated ML models.
> > >
> > >
> > > **The study of the multi-task setting is where the paper best distinguishes itself from related work, and in my view this setting would be worth expanding, while prior sections can be condensed.**
> > >
> > > We agree that the multi-task setting of section4 is the most interesting and relevant, while the Saxe et al re-analysis from section 3 is primarily meant to be an extended introduction to the eNTK. We have condensed this section, removing experiments that are not relevant to section 4 (as well as a redundant analysis) to the appendix, while also shortening the eNTK section by removing the ePNTK elements (which were not used in section 4) to the appendix.

---

> > > > ### Comment · Reviewer_P5dv · 2023-10-13
> > > > **Thanks for the paper updates, substantive concerns remain (Pt 1)**
> > > >
> > > > Thanks to the authors for their updates to the paper. The plots are now much more readable and the eNTK background is now much clearer. While many of the object-level concerns that I raised in my original review have been addressed, the higher-level issues concerning clarity and significance to the TMLR readership remain. I would like to highlight these to give a chance to respond before I submit my final evaluation.
> > > >
> > > > To start, I think there is still some confusion over what the primary contribution of the paper is. I appreciate the authors' clarification that "the main goal of this article is to use the eNTK to analyze shared vs separated representations [...] Primarily, we wanted to introduce and study the notion of simultaneous multitasking [...] to the machine learning community [...]".  I think this is certainly an interesting idea. However, looking at the allocation of space in the paper, I was not under the impression that this was its main goal; rather, I'd thought that the paper was about developing a set of analysis methods, which had the side effect of being applicable to the simultaneous multitasking setting. If the quoted statement does indeed characterize the intended primary contribution of the paper, then I think the article would need to be extensively rewritten, to include a separate section outlining background and **motivating** the study of this simultaenous multitasking problem, and highlighting example application areas where it would be of interest to the machine learning community. I think motivation in particular is lacking in the current version of the paper: the current strategy of concatenating inputs to the network as opposed to batching increases computational cost of inference on a single example (due to the growth in the number of parameters from the increased input size along with the likely necessary increase in the hidden dimension) and introduces the risk of the network overfitting to this fixed number of concatenated inputs. In this case, the lengthy introduction of analysis techniques could largely be covered by citations of prior work studying the empirical NTK, giving more room for a richer experimental analysis of variations on the current simultaneous multitask setup in the paper.
> > > >
> > > > If, however, the paper is intended as primarily a methods paper, I believe it requires significant additional work to distinguish it better from the existing literature. While I highlighted a handful of obvious missing references in my original review of all instances where I thought the contribution of the paper exhibited significant overlap with existing work or was unclear, I will provide additional examples now which stood out on a read-through of the updated manuscript.
> > > >
> > > > As a first example, the introduction states "In this article, we expand upon the ideas introduced in Sahs et al. (2022) to show how the initial condition of a network and the training regime to which it will be subjected can predict the implicit bias and thus the kinds of representations it will learn[...] This offers a **new method** to analyze how networks can be optimized to regulate the balance between flexibility and efficiency." Having read the paper, I still don't feel that I understand what this "new method" is -- is it the M-PHATE visualizations, the eNTK clustering, or the correspondence between the two? The first and last of these possibilities require training the network, which seems to defeat the purpose of studying a property only of the initialization. Similar analysis of the eNTK has previously been conducted in several prior works; for example Fort et al. (2020) study the evolution of the eNTK over time, . Further, other properties of a random initialization have also been conducted to relate expressivity to initialization scale, see e.g. Raghu et al. (2017).

---

> > > > > ### Comment · Reviewer_P5dv · 2023-10-13
> > > > > **Response (pt 2)**
> > > > >
> > > > > Second, many of the claims in section 3 are unclear or do not properly cite prior work which performed similar analyses. This gives the impression that the analytical tools in this section are new, when in fact they closely resemble the methods used in related works. In particular:
> > > > >
> > > > > "The eNTK allows us to examine the patterns that are being learned at each time point, by examining the eigenvectors of the eNTK."
> > > > >
> > > > > -> Eigenvector analysis of the empirical neural tangent kernel has been conducted previously in the literature, most notably by Loo et al. (2020) who provide quite striking visualizations of the empirical ntk in convolutional networks.
> > > > >
> > > > > "These results are comparable to those of the analysis of the linear system (Fig 3), providing support for the generality of eNTK analyses to non-linear networks."
> > > > >     -> Both clauses of this sentence are not clear. First, it is not clear what "analysis of the linear system" means so it's difficult to understand what the eNTK is being compared against. Second, it is known that the empirical neural tangent kernel can accurately predict training dynamics in some learning problems, and in other cases fail to accurately predict highly nonlinear training dynamics, for which plenty of examples exist in the literature (e.g. Fort et al. (2020), Lewkowycz et al. (2020), Atanasov et al. (2022)). Given this context, I fail to see the utility to the broader field in providing an example of a two-layer network where the eNTK is an effective predictor of the training dynamics.
> > > > >
> > > > > Unclear or overexaggerated claims:
> > > > > "It is important to emphasize that the eNTK analysis is predictive, in that the results are derived only from the conditions of the network at the time point to which the analysis is applied, but predict the representational organization of the network following subsequent learning given the training regime."
> > > > >
> > > > > -> I couldn't figure out which figure was intended to support this claim -- how are we supposed to predict long-range shapes of the curves in Figure 4 from their initial values?
> > > > >
> > > > > "...against the projections of the data x into the coordinate system spanned by the singular vectors of Wtarget. This allows us to compare how much each data point contributes to the eNTK compared to how much it would contribute if it was perfectly learning the modes of Wtarget, an approach that works for linear or nonlinear systems."
> > > > >
> > > > > -> This sentence states that analysis can generalize to nonlinear learners, but it seems like it is still limited to linear targets (most interesting problems are not linear functions, and so it seems like a bit of an overstep to claim that this is supporting the generality of the analysis).

---

> ### Comment · Reviewer_P5dv · 2023-10-13
> **Response (pt 3)**
>
> There are similar issues in the later sections as well, though providing a line-by-line catalogue would make this an excessively long review both for the reader and the writer. The biggest object-level issues that I found and which I would want to see addressed before recommending acceptance are the following:
>
> 1) Section 4 studies the eNTK of the network architecture described in the concurrent-multitasking problem, and claims that the clustering of inputs by the NTK eigenvectors early in training is predictive of the feature geometry evolution as characterized by M-PHATE. I think this claim would be much stronger if the results were replicated in more architectures which induce a more nonlinear loss landscape. For example, if we replaced the single hidden layer in the blue square with a deeper or more expressive network, would the same results hold? Or is the interference between tasks limited to networks with limited expressivity?
>
> 2) Section 5 makes the claim that "the eNTK analysis, carried out prior to any training, is able to predict the kinds of representations learned by the network in response to different initializations, but also that it can be used to predict the patterns of generalization performance associated with those representations observed in the network after two distinct phases of training." For this claim to be of interest to the TMLR readership, I think it would need to be made with greater generality. Right now it is only justified as applied to a single, somewhat contrived network architecture.
>
> 3) I was not aware of PHATE or M-PHATE prior to reading this paper, and given the importance of Figure 7 to the narrative around shared representations, I think it would be beneficial to include an expanded description in Section 2 of what it is computing and thus what the visualizations signify.
>
> 4) Minor nit: many of the figures take up an entire page when they do not need to do so (e.g. Figures 2, 4, and 10).
>
> In spite of the issues raised above, I do think that this paper has the potential be quite interesting and insightful. Studying the learning dynamics of neural network models of simultaneous multitasking is a fresh idea, and it would be great to see it provide new insights that could be used by the psychology community. But in order for me to recommend that this paper, I would need to see the above comments materially addressed, rather than listed as future work as is done in the current manuscript.

---

> > ### Author Response · Authors · 2023-11-08
> > **Final Response to Reviewer 2 (P5dv)**
> >
> > We would like to thank the reviewer for their additional comments. We would like to address  our responses to the following main concerns:
> >
> > **Novelty of the eNTK**
> >
> > The reviewer brought up previous work, including computational evaluations of finite-width NTKs, and questioned the unique components of the eNTK relative to these previous works. Although the eNTK is closely related to the NTK, they do measure different properties of the network. We have made further revisions to the eNTK section in order to make the difference more clear - this includes fixing a mistake that may have resulted in some of the confusion.
> >
> > In addition we have added a new section in the appendix that directly compares and contrasts an eNTK analysis to an analysis done using a computational evaluation of the NTK, while providing additional context and intuition to the difference between the eNTK and NTK.
> >
> > **Relation to Previous Work**
> >
> > The reviewer brought up several previous works of interest. After reviewing, we found Fort et al. (2020), Lewkowycz et al. (2020), and Loo et al. (2022) to be particularly relevant. Fort and Lewkowycz reviewed learning dynamics and loss landscapes, finding that learning was often highly adaptive early on, with NTKs shifting during learning, but that past a certain point NTKs stabilized and kernel analysis methods could predict the rest of the learning process. In contrast, Loo et al. (2022) reviewed adversarial robustness training, including an interest experiment where they showed that NTK eigenvalues in adversarially-robust networks were more interpretable.
> >
> > We have referenced these papers in the main text when first introducing the NTK, making it explicit that computational work with finite-width NTKs has led to interesting prior work.
> >
> > **Minor Issues Addressed:**
> >
> > We took the reviewer’s advice, and tightened up many figures. This moderately reduced paper length.
> >
> > We make it clearer that the eNTK analysis is predictive to a useful extent in the network we study (simultaneous-multi tasking network). In general we expect the eNTK to remain predictive, although the time horizon may vary to the extent that predictions are not practically useful. We briefly discuss this in the conclusions and future work section
> >
> > We rework the description of the analysis of the experiment of figure 3, making it clear that we are comparing the exact ground-truth (available due to analyzing a linear system) to the eNTK analysis (a general technique usable in linear or non-linear systems).
> >
> > When we show that the eNTK is predictive for the SMG system, we are primarily referencing figure 5, which shows that the active learning singular vector reaches full agreement with the specified target singular vector BEFORE loss begins to drop as the mode is learned. For example, singular vector 2 becomes the active singular vector before t = 500, while the second loss drop does not begin until around t = 600.
> >
> > We expand upon the description of the M-PHATE and PHATE decomposition techniques, providing a high level overview of the underlying PHATE method, before explaining how it is used by M-PHATE.

---

### Review · Reviewer_Qhfk · 2023-09-11

**Summary Of Contributions:**

This paper proposes a technique for analyzing the training dynamics of neural networks. Specifically, they introduce a kernel-based analysis (neural tangent kernel, or NTK) which allows predicting the future behavior of the network. The paper validates the NTK analysis on the behavior of a simple linear network that had been analyzed in prior works, and then uses it to analyze more complex networks in the context of multitask learning. The paper concludes that NTK can be a useful analysis for predicting how a network's behavior will evolve over time.

**Audience:**

Yes

**Broader Impact Concerns:**

No concerns.

**Claims And Evidence:**

No

**Requested Changes:**

To improve clarity, one general suggestion would be to be a little bit more careful about notation, and be more explicit about how things are described. I make some specific suggestions below. However, more broadly, I came away from reading the paper with a lot of confusion about how to actually apply the method. Could the authors provide an algorithm box, or additional section explaining more precisely how the method should be used? Specifically: what gets computed, and then how should that number be interpreted/analyzed?

I also found many of the figures hard to understand. For example in Figure 4 it's not at all obvious to me why the top left shows the first mode is being learned while the top right shows it has already been incorporated (those subplots look identical to me so it's not clear what's the difference?). In Figure 9, I had to stare at the plots for a long time to notice the differences. This is a problem, because if the differences in the analysis are not easy to see, that does not do much to convince me that the analysis is useful.

Specific comments / points of confusion:
- Please add equation numbers to the equations in Section 2, to make them easier to reference.
- $L$ in the first equation of section 2.1.1 is not defined (I assume this is the loss, but would be good to be explicit).
- The dot notation for time derivatives is inconsistent with the notation for the other derivatives. I think it would be easier to follow if the notations were not mixed (i.e., explicitly write $\frac{d\theta}{dt}$ rather than $\dot{\theta}$).
- It would help to be explicit about what the second equation is. i.e., it is how the network parameters change over time.
- What happened to the minus sign in the third equation of Section 2?
- So, $\epsilon_m$ is the same as $\frac{dL}{dy}$? This should be stated explicitly. I am not sure introducing the additional term "loss sensitivity" helps, especially as it's not explained what it means.
- $\phi_m$ is defined as the "vector of NTK" features. The NTK hasn't been introduced yet, so this is confusing---I don't know what "NTK features" means.
- It would help to be explicit about what the fourth equation is. i.e., it is how the NN predictions change over time.
- It would help to be explicit about what the fifth equation is. i.e., it is how the gradients of the NN loss change over time.
- I got a bit lost with the derivation of the fifth equation. I think it would help to go through it a little more carefully and be more explicit.
- To be honest, when the paper says it is using the NTK analysis, I don't actually know what is being computed. Is it $\dot{\epsilon}_n(t)$ (fifth equation)? Or something else? And how is that evaluated at time t>0 as we don't actually know what the weights are at that point in time? I think it could be very helpful to the reader to specifically walk them through the algorithm by which you'd apply the NTK analysis.
- Page 4: "the NTK will not change much during training" - can you explain why this is? It wasn't obvious to me.
- Page 6 equations: how are these computed in practice? Can the integrals be computed analytically (in which case please provide the analytic form) or do they need to be approximated? If approximated, how do you do this in practice without actually computing the loss at different points in time?
- Please make the font size larger in all the figures (it should be roughly the same size as the text)

**Strengths And Weaknesses:**

Strengths:
- important topic (analysis and interpretability of NNs)
- nice connections to the neuroscience literature
- replication of results in prior literature

Cons:
- difficult to understand
- unclear how to apply the method to a new setting

I will begin my review by prefacing that this is not my research area, and so I am reviewing the paper more generally from the point of view from someone in machine learning who would be interested in using the proposed analysis to understand NN behavior. So it is possible that I am missing some context of related research and what is standard practice for NN analysis, and therefore my review should be taken with a grain of salt.

That said, I found this paper quite difficult to grasp. While I get the high level motivation and idea, I don't really understand everything about how the method is derived or how I would use it to analyze the behavior of a network not described in the paper. I think that this method could be potentially very useful but given my confusion I am unable to assess that.

I have an additional concern about the computational complexity of the method. I might be wrong given that I didn't fully understand how it works, but it seems to be that (especially for computing the integrals of the PNTK) you would need to compute the model gradients for every data point in the training set or every point during training. If this is the case, what does the method buy you over simply training the network and empirically measuring the behavior? Perhaps I am missing something, and it would be helpful if the authors could clarify.

---

> ### Author Response · Authors · 2023-09-23
> **Response to Reviewer 1 (Qhfk) (1/2)**
>
> We would like to thank the reviewer for their helpful comments, and are glad to hear that they are interested in using eNTK analyses in their own research. We addressed their following comments and requests:
>
> **Difficult to Understand:**
>
> We reworked the eNTK section, adding new intuition, enhanced explanations, standardized mathematics, and removed secondary analyses to the appendix. We hope this makes the eNTK section significantly clearer.
>
> **Unclear how to apply the method to a new setting:**
>
> We added new detail to the eNTK which should make explicit how we calculate the eNTK in practice. We also added a new discussion section that goes into the potential benefits and pitfalls of using eNTK on larger, more complicated networks.
>
> **I will begin my review by prefacing that this is not my research area, and so I am reviewing the paper more generally from the point of view from someone in machine learning who would be interested in using the proposed analysis to understand NN behavior. So it is possible that I am missing some context of related research and what is standard practice for NN analysis, and therefore my review should be taken with a grain of salt. That said, I found this paper quite difficult to grasp. While I get the high level motivation and idea, I don't really understand everything about how the method is derived or how I would use it to analyze the behavior of a network not described in the paper. I think that this method could be potentially very useful but given my confusion I am unable to assess that.
> I have an additional concern about the computational complexity of the method. I might be wrong given that I didn't fully understand how it works, but it seems to be that (especially for computing the integrals of the PNTK) you would need to compute the model gradients for every data point in the training set or every point during training. If this is the case, what does the method buy you over simply training the network and empirically measuring the behavior? Perhaps I am missing something, and it would be helpful if the authors could clarify.**
>
> The ePNTK is a secondary analysis which, due to its computational cost, we did not actually use in our main analysis on the shared vs separated representations task. The ePNTK section was moved to the appendix, with a note on its cost. Additionally, it received the same treatment as the eNTK section in order to standardize notation and make the logic clearer.
>
> **To improve clarity, one general suggestion would be to be a little bit more careful about notation, and be more explicit about how things are described. I make some specific suggestions below. However, more broadly, I came away from reading the paper with a lot of confusion about how to actually apply the method. Could the authors provide an algorithm box, or additional section explaining more precisely how the method should be used? Specifically: what gets computed, and then how should that number be interpreted/analyzed?**
>
> We added a notation paragraph which lays out our (newly standardized) mathematical notation for the eNTK (and ePNTK). We also expanded the final eNTK discussion section, adding additional details about how the eNTK is computed, and how it can be interpreted and analyzed.
>
> **I also found many of the figures hard to understand. For example in Figure 4 it's not at all obvious to me why the top left shows the first mode is being learned while the top right shows it has already been incorporated (those subplots look identical to me so it's not clear what's the difference?). In Figure 9, I had to stare at the plots for a long time to notice the differences. This is a problem, because if the differences in the analysis are not easy to see, that does not do much to convince me that the analysis is useful.**
>
> All figures had their caption changed in order to make the main point explicitly clear after the figure description. Where appropriate, figures had their titles, labels, and legends updated. Additionally, moving the ePNTK analysis to the conclusion makes some of the figures simpler and clearer, as they now only feature the eNTK results (such as the mentioned figure 4). Figure 9 was also moved to the appendix, with its place in the main text replaced with a new figure,  which should distill the main results from the original figure 9 in a more intuitive format.
>
> **Please add equation numbers to the equations in Section 2, to make them easier to reference.**
>
> Equation numbers (and references to them) added.
>
> **L in the first equation of section 2.1.1 is not defined (I assume this is the loss, but would be good to be explicit).**
>
> L (and other variables) defined explicitly in the new notation section.

---

> > ### Author Response · Authors · 2023-09-23
> > **Response to Reviewer 1 (Qhfk) (2/2)**
> >
> > **The dot notation for time derivatives is inconsistent with the notation for the other derivatives. I think it would be easier to follow if the notations were not mixed (i.e., explicitly write dl/dtheta rather than dot(theta)**
> >
> > We standardized derivative notation to follow the suggested format.
> >
> > **It would help to be explicit about what the second equation is. i.e., it is how the network parameters change over time.**
> >
> > We added a small passage clarifying this.
> >
> > **What happened to the minus sign in the third equation of Section 2?**
> >
> > We thank the reviewer for catching this mistake, which has been corrected.
> >
> > **So, epsilon is the same as dL/dy? This should be stated explicitly. I am not sure introducing the additional term "loss sensitivity" helps, especially as it's not explained what it means.**
> >
> > The loss sensitivity is explicitly defined. The overall goal of the loss sensitivity term is to make the final kernel form clearer, as the loss sensitivity is the kernel weight.
> >
> > **phi is defined as the "vector of NTK" features. The NTK hasn't been introduced yet, so this is confusing---I don't know what "NTK features" means.**
> >
> > As part of the clarity rewrite, the feature vector is introduced first, then clarified to be the eNTK feature vector after the eNTK is formalized.
> >
> > **It would help to be explicit about what the fourth equation is. i.e., it is how the NN predictions change over time.**
> >
> > We introduce the yhat notation in the notation section, and also clarify the equation meaning.
> >
> > **It would help to be explicit about what the fifth equation is. i.e., it is how the gradients of the NN loss change over time. I got a bit lost with the derivation of the fifth equation. I think it would help to go through it a little more carefully and be more explicit.**
> >
> > We reworked this section to be clearer. Equation 5 now directly references the output, rather than the residual, and its derivation from equation 4 should be clear.
> >
> > **To be honest, when the paper says it is using the NTK analysis, I don't actually know what is being computed. Is it epsilon (fifth equation)? Or something else? And how is that evaluated at time t>0 as we don't actually know what the weights are at that point in time? I think it could be very helpful to the reader to specifically walk them through the algorithm by which you'd apply the NTK analysis.**
> >
> > We more explicitly define what the eNTK is (the kernel), and that it is computed from equation 6, while the overall change in the test outputs using the eNTK is computed from equation 6.
> >
> > **Page 4: "the NTK will not change much during training" - can you explain why this is? It wasn't obvious to me.**
> >
> > This is true by definition in the kernel regime, which we add a note to remind the reader. Intuitively, in the kernel regime the NN uses a set of fixed features, which are exactly the eNTK feature vectors.
> >
> > **Page 6 equations: how are these computed in practice? Can the integrals be computed analytically (in which case please provide the analytic form) or do they need to be approximated? If approximated, how do you do this in practice without actually computing the loss at different points in time?**
> >
> > We make clear that the ePNTK is also empirically estimated, and provide additional details. We also have moved this section to the appendix, as it is less relevant to the rest of the work.
> >
> > **Please make the font size larger in all the figures (it should be roughly the same size as the text)**
> >
> > We have gone through all figures and increased various font sizes as necessary.

---

> ### Comment · Reviewer_Qhfk · 2023-10-18
> **Response to authors**
>
> I appreciate the extensive revisions of the article---I find it much clearer to read and understand, and most of my original points of confusion have been addressed. Thank you very much for the work you put into doing this!
>
> However, in reading through the revised article there were a few points that still stood out to me, related to things I mentioned before but which I can see much more clearly now:
>
> 1. Computational cost: Section 6 states that "The eNTK computation is tractable, costing only slightly more than an epoch of training. Thus, we expect the eNTK will scale to arbitrary network architectures and sizes." I appreciate that this assumption is now more clearly stated. However, the "single epoch" point seems like a strong assumption given that most modern neural networks use only one or maybe two epochs of training, or don't even have a fixed dataset size at all (in the case of RL). This means that eNTK is limited to supervised learning scenarios where dataset sizes are tiny, which is typically not the setting where neural networks are actually used. Perhaps this is an acceptable limitation for a cogsci/neuroscience audience, but for machine learning more generally it seems like a major limitation to me that would greatly decrease potential interest of the TMLR audience.
>
> 2. Applicability to new tasks: I wrote in my initial review that "I came away from reading the paper with a lot of confusion about how to actually apply the method". While the specific details of this have been clarified, I still find it unclear how exactly this analysis would be best applied to arbitrary new settings. In the multi-task case study from the paper, the analyses rely on being able to perform averages across higher-level statistics (e.g. input groups and task groups). What if I don't have something like this? Say I construct a new variation of a transformer architecture and I want to train it to predict the dynamics of a physical system, and prior to training I want to know which architectural choices will perform best (e.g. number of layers). How can I used eNTK to help me with this? Is there an implicit assumption that the task must be classification, so that you can predict something about the categories on average? I think it would help to provide more examples of how eNTK can be used for a range of different tasks and setups. In particular, if you could address this by applying eNTK to more cogsci/neuro-inspired tasks this could also help to address point #1 above, by at least making it clear that this would be broad interest and applicability to those fields.
>
> 3. Finally, I have to disagree with the authors in their rebuttal regarding the length of the paper. While length was not something I mentioned initially, I agree that the paper is quite long and at times it is overly verbose, which makes it hard to read and to quickly understand the main points. As an example, Section 4.1.1 on the network architecture is almost a full page long when it could be stated in a couple of sentences in the main text, with details relegated to the appendix. (For example: "The architecture is a 1-layer MLP with weights $W$ and $V$ for the hidden and output layers, respectively, and sigmoid activations on both layers. The inputs to the first layer are binary stimulus features $x_1$ and binary task features $x_2$. There are $g_1$ stimulus dimensions each represented as an $m$-dimensional one-hot vector, for a total of $|x_1|=g_1\cdot{}m$ binary stimulus features. The inputs to the second layer consist of the hidden activations from the first layer concatenated with $x_2$."). While this is one example, there are many such places in which the text is overly verbose, and by addressing this the manuscript could be reduced to 1/2-2/3 the length without needing to actually cut any of the content.
>
> Unfortunately, I believe that addressing the points above---and in particular #2---would still require substantial revisions before I would be happy recommending acceptance.

---

> > ### Author Response · Authors · 2023-11-08
> > **Final Response to Reviewer 1 (Qhfk)**
> >
> > We would like to thank the reviewer for their additional comments. We would like to address  our responses to the following main concerns:
> >
> > **Computational Cost**
> >
> > It is true that the eNTK analysis of an epoch is more expensive than a standard training of that epoch. Thus, for use cases with very few, very expensive epochs (such as Large Language Models), the eNTK may be prohibitively expensive. It also means that the eNTK is not well suited to analysis of streaming data tasks, such as Reinforcement Learning. Nevertheless, we believe that this leaves numerous applications where the eNTK may still be applied, including image analysis, LLM fine-tuning, and bespoke NN models such as the multi-tasking model analyzed in this work. In addition, it is possible to compute either a partial eNTK (only analyzing influences from a certain subset of training data of interest), or a grouped eNTK (analyzing influences combined into groups such as batches), both of which can significantly reduce the storage and compute requirements.
> >
> > The conclusions section that discusses eNTK cost and scaling has been updated to include some of the above discussion
> >
> > **Applicability to New Tasks**
> >
> > The eNTK analysis used here relied heavily on our ability to group over input and task settings. In general, we expect this technique to continue to be useful - the eNTK is a data-centric approach, so grouping of properties of the inputs (which here included task information) or outputs will continue to make sense. For example, in an image classification task it might make sense to group by class, while for an LLM groupings could include sentiment, data source, or genre. This can be used to check specific hypotheses (as was done in this work), or be used as an exploratory aide (by e.g. checking correlations over various properties). Although presented on a classification task, the eNTK could also be used on any type of outputs, including regression or generative tasks. As a data-centric approach, the eNTK doesn’t directly provide architectural information (such as the correct number of layers in a new transformer subtype), although it could be used to analyze individual networks for A/B testing across architectures.
> >
> > The conclusion section that discusses the eNTK has been expanded with a new paragraph that includes some of the above discussion.
> >
> > **Length**
> >
> > We agree that the paper is quite lengthy, and has some extraneous details that can be moved to the appendix and over-large figures. We have made another pass through, moving architecture and task details (we appreciate this suggestion) to the appendix while leaving summaries in the main text, as well as tightening up figures. Length continues to be challenging, as we have received conflicting asks for increased clarity and exposition and reduced length, but we believe we have condensed the main details as much as is practical.
> >
> > Overall changes in this revision, despite clarifying additions to related work, experimental interpretations, and expanded eNTK conclusion discussion, have reduced length by 2 pages.

---

### Author Response · Authors · 2023-09-23
**Summary of Changes**

We would like to thank the reviewers for their useful feedback, particularly with regard to clarity issues. We wish to go over the main points of reviewer feedback, and our main changes in response:

## Clarity was brought up by all reviewers, particularly with respect to the figures and the NTK methodology. In order to address this, we:

- Reworked the captions of all figures to explicitly include the main point. Figures’ font sizes were also generally increased, and axis labels / titles / legends were reworked in order to increase readability or clarity.
- Renamed the NTK to the empirical NTK (eNTK) in order to highlight the difference between our proposed method and previous theoretical NTK results. We also added a short section explicitly referencing this difference.
- Added more explicit information about how the eNTK is computed
- Reworked variables and equations to be more standardized and explanatory. We also added a short notation section to introduce readers to the important terms.
- Moved some secondary analysis and results, including all ePNTK results, to the appendix. These were more complex and primarily of theoretical interest, as no ePNTK results were used during our main analysis of the shared vs separated representations task.
- Moved (old) Figure 3 and related experiment to the appendix. This was a complementary, alternate method to figure 4, but more complicated to explain.

## Relations to existing work. In order to address this, we:

- Added a new section to the eNTK in order to better differentiate our proposal from previous, theoretical NTK results
- Added in a new section tying our method of modifying initialization to control Inductive Bias (IB) to previous works, including most suggested by reviewers.

## Length. Our paper is longer than the TMLR average. We took several steps to shorten our paper:

- Moved redundant eNTK analysis, all ePNTK introductions, explanations, and experiments to appendix.
- Reshaped several figures to condense the amount of information into a smaller space
- Reworked a particularly large and complicated figure (originally Figure 9) using a summary statistic, and moved the original figure and analysis to the appendix.

## How our eNTK methodology can be used on other tasks, particularly those with higher complexity or scale

- We clarified how the eNTK is computed and used in practice when in the eNTK introductory section
- We added a new section to the conclusion which expands upon the utility and possible pitfalls of using the eNTK on larger or more complicated architectures
- We proposed future work to use the eNTK to analyze a more complicated psychology-inspired neural network model which has a more realistic (non-binary) model of shared vs separated representations.

## Overall, we feel that these preceding changes cover a large majority of the reviewers requested changes. There are two remaining considerations we wish to cover in more detail.

The first is the length of our paper, which is still long even after the above reductions. Our paper is indeed longer than average TMLR submission, but we feel that this is appropriate given the context. This work presents a novel technique, introducing and explaining it on a simple example, before using it to analyze our main shared vs separate representations task. This task is directly  motivated by psychology literature and results, requiring an expository introduction. As we are dealing with a novel combination of a new technique applied to an interdisciplinary problem, and given the need for clarity in bringing these aspects together, we do not feel that it is possible to make further reductions in length.

The second is the relative simplicity of the network model we use to examine the role of shared vs separated representations, both in terms of machine learning considerations like number of parameters, as well as in terms of psychological assumptions about the complexity of stimuli. Since even this simplified setup required significant exposition, we left a more complicated psychology setup for future work. However, the main goal of this article is to use the eNTK to analyze shared vs separated representations, rather than benchmarking on a standard large-scale ML task. Primarily, we wanted to introduce and study the notion of simultaneous multitasking (as opposed to multi-tasking e.g. ability to perform one of multiple tasks) and related ideas of cognitive control from the psychology literature to the machine learning community, demonstrating the key issue with a minimal example. We believe this to goal to be relevant to TMLR’s scope, which calls for papers that contain:
- experimental and/or theoretical studies yielding new insight into the design and behavior of learning in intelligent systems;
- computational models of natural learning systems at the behavioral or neural level;
- new approaches for analysis, visualization, and understanding of artificial or biological learning systems.

---

### Author Response · Authors · 2023-11-08
**Final Response to Reviewers**

We would like to thank the reviewers for their additional comments. We have made another round of revisions in response to reviewer’s suggestions. We highlight some of the most salient comments, and our changes:

**Computational Cost**

The eNTK analysis of an epoch is more expensive than a standard training of that epoch. Thus, for use cases with few, very expensive epochs (such as Large Language Models, or the effective single epoch of RL), the eNTK may be prohibitively expensive. Nevertheless, we believe that this leaves numerous applications where the eNTK may still be applied, including image analysis, LLM fine-tuning, and bespoke NN models such as the multi-tasking model analyzed in this work. In addition, it is possible to compute either a partial eNTK (only analyzing influences from a certain subset of training data of interest), or a grouped eNTK (analyzing influences combined into groups such as batches), both of which can significantly reduce the storage and compute requirements.

The conclusions section that discusses eNTK cost and scaling has been updated to include some of the above discussion

**Applicability to New Tasks**

The eNTK analysis used here relied heavily on our ability to group over input and task settings. In general, we expect this technique to continue to be useful - the eNTK is a data-centric approach, so grouping of properties of the inputs (which here included task information) or outputs will continue to make sense. This can be used to check specific hypotheses (as was done in this work), or be used as an exploratory aide (by e.g. checking correlations over various properties). Although presented on a classification task, the eNTK could also be used on any type of outputs, including regression or generative tasks.

The conclusion section that discusses the eNTK has been expanded with a new paragraph that includes some of the above discussion.

**Length**

Reviewers have continued to note the length of the paper, in particular with extraneous details that can be moved to the appendix and over-large figures. We have made another pass through, moving architecture and task details to the appendix while leaving summaries in the main text, as well as tightening up figures. Length continues to be challenging, as we have received conflicting asks for increased clarity and exposition and reduced length, but we believe we have condensed the main details as much as is practical.

Overall changes in this revision, despite clarifying additions to related work, experimental interpretations, and expanded eNTK conclusion discussion, have reduced length by 2 pages.

**Novelty of the eNTK**

Reviewers have brought up previous work, including computational evaluations of finite-width NTKs, and questioned the unique components of the eNTK relative to these previous works. Although the eNTK is closely related to the NTK, they do measure different properties of the network. We have made further revisions to the eNTK section in order to make the difference more clear - this includes fixing a mistake that may have resulted in some of the confusion.

In addition we have added a new section in the appendix that directly compares and contrasts an eNTK analysis to an analysis done using a computational evaluation of the NTK, while providing additional context and intuition to the difference between the eNTK and NTK.

---

### Decision · Action_Editor_Pr81 · 2023-11-22

**Recommendation:** Reject

**Comment:**

I have just re-read the latest version of the manuscript and all of the reviewer-author interaction again. I do believe that the first round of reviews has led to significant improvements to the paper in terms of clarity and distinguishing eNTK from other NTK-based work. I want to thank the authors for their effort, responsiveness, and polite and constructive engagement with the reviewers throughout. Having said that, all reviewers have raised similar issues after the first round of improvements, which I consider partly, but not fully, addressed with the second round of improvements (and having such a lengthy back-and-forth without a clear outcome in favor of accepting the manuscript might also be quite indicative of a more substantial issue remaining). To me, the current paper is attempting to do the following 3 things - my personal recommendation is to write a concise and convincing paper for either one of these points but not more than one:
  * Introduction of eNTK as a method. This is interesting but unfortunately the paper only demonstrates this in a very limited setting, and some concerns with more complex tasks and network architectures are already foreseeable but neither empirically investigated nor fully addressed (but they do appear in the discussion). This could be developed into a whole separate methodology paper, but that paper would need more comparison against modern empirical uses of NTK (why eNTK over other methods, are there systematic advantages / shortcomings between the different options, ablation studies, more complex settings, empirical comparisons with alternatives, etc).
 * Combination of eNTK and M-PHATE to qualitatively analyze internal representations of networks. This could be interesting to the neuroscientific / psychological community and the interpretability community. To develop this into a paper it would need to be strengthened  how/why eNTK contributes over simply studying representations with M-PHATE  (eNTK is currently shown to be predictive of one aspect of learned representations; what other aspects could it predict reliably?). To show relevance to the neurocientific community some experiments/visualizations on neural recordings would be highly interesting; for machine learners more complex tasks and larger neural networks would be interesting.
 * Investigation into how the initialization of artificial neural networks leads to shared vs. distinct representations. This is another very interesting question, but not entirely novel in itself. To develop this into a paper the question is what does eNTK/M-PHATE analysis contribute? What general findings can be taken away (again hard to argue from a very simple toy example only); especially given the intro of the manuscript which is strongly motivated by (often behavioral) findings in neuroscience/psychology. It is unclear per-se how relevant findings on a highly simplified neural network and an extremely simplified task have are to biological neural networks in complex sensorimotor tasks. Bridging this gap is difficult and may not work out.

The verbosity of the manuscript, lack of focus and at least some confusion around the motivation and main contributions, have been pointed out by all three reviewers - and while the authors have put in considerable effort to address this, I think the current manuscript is currently too verbose and not sharp/deep enough at the same time. A significantly shortened manuscript with clearly stated claims (of a more limited extent), clearly stated main contributions and novel parts, and more discussion of current limitations could pass the bar for TMLR. Unfortunately I do not think we have reached this state yet given the two rounds of improvements. I therefore suggest a major revision (which is equivalent to a Reject at TMLR, but with the option to resubmit at a later stage). With the paper as is, perhaps a slight revision to target a psychology audience is also possible - which would mean resubmission to another outlet. I personally would want to encourage the authors to put in some more effort into this project (which certainly shows some interesting early results) and turn the manuscript into an impactful paper. I do want to thank the authors again for their engagement and work; I think the manuscript was significantly improved, and I understand that this outcome is frustrating. I personally would actually be quite looking forward to reading one of the three "simplified and more targeted" versions of the work as I outlined them above.

**Audience:**

I do not believe that the paper is currently in the completely wrong venue (though perhaps addressing a somewhat small sub-community within the TMLR audience). I do believe, however, that the current paper attempts to primarily address 3 (somewhat distinct) audiences at the same time (see my comments below). This leads to an overly lengthy paper that simultaneously lacks depth for each of the three potential target audiences. I personally recommend strengthening the focus and potentially splitting the manuscript into two papers.

**Claims And Evidence:**

The paper introduces eNTK, a novel analysis technique that is shown to be predictive of whether a single-layer MLP learns a geometrically  non-trivially structured representation of the inputs or not. The evaluation uses a multi-task setup where, across tasks, disjoint groups of input neurons are fully predictive of disjoint groups of output neurons (the analogy is that each input group could be e.g. a different modality). The advantage of eNTK is that it is sensitive to a network's initial weights and predictive of the geometric structure of the learned representation after one epoch of training - at least for a static dataset. While eNTK works well in the toy example shown, the paper points out some potential challenges and foreseeable shortcomings in more complex settings, but does evaluate empirical performance for more complex tasks or neural networks. This means the claims around eNTK made in the paper are not technically wrong, but I consider them overstated given the abstract and intro - the limitations are only briefly addressed in the Discussion and Conclusions Section.

Besides introducing eNTK, the paper claims to study and answer under which conditions neural networks learn shared vs distinct representations in a multi-task setting with shared structure between tasks. While these notions are never made formal, the paper uses a very simple example to empirically/ad-hoc define what "shared task structure" is and uses qualitative visualizations to define what constitutes a "structured representation". eNTK performs well on this toy example, and is predictive (after one epoch of training) of whether a structured hidden representation is formed or not - which is also predictive of downstream performance when fine-tuning the trained network to perform multiple tasks simultaneously. Similar to the point above, if the claims in the paper were reduced to the very limited setting studied (and reduced to being able to predict one aspect about the structure of learned representations), I would be happy. Unfortunately, the abstract and intro (and some parts deeper into the paper) make much wider claims. While these claims may hold in practice, the current empirical evaluation simply cannot support very strong claims.

The paper would strongly benefit from more toned-down, realistic claims (more forward looking claims can still be put into the discussion as future ambitions). Additionally, the paper would benefit from very clearly stated contributions (including their scope, and which parts of the paper are novel) at the end of the introduction.

**Resubmission Of Major Revision:**

The authors may consider submitting a major revision at a later time.